# WorkArena++: Towards Compositional Planning and Reasoning-based Common Knowledge Work Tasks

**Léo Boisvert**[*123]   **Megh Thakkar**[*124]  **Maxime Gasse**[†123]  **Massimo Caccia**[†1]
**Thibault Le Sellier De Chezelles**[†13]  **Quentin Cappart**[23]  **Nicolas Chapados**[123]
**Alexandre Lacoste**[‡1]  **Alexandre Drouin**[‡12]
[1]ServiceNow Research, [2]Mila, [3]Polytechnique Montréal, [4]Chandar Research Lab
{leo.boisvert, megh.thakkar, alexandre.drouin}@servicenow.com

## Abstract

The ability of large language models (LLMs) to mimic human-like intelligence has led to a surge in LLM-based autonomous agents. Though recent LLMs seem capable of planning and reasoning given user instructions, their effectiveness in applying these capabilities for autonomous task solving remains underexplored. This is especially true in enterprise settings, where automated agents hold the promise of a high impact. To fill this gap, we propose WorkArena++, a novel benchmark consisting of 682 tasks corresponding to realistic workflows routinely performed by knowledge workers. WorkArena++ is designed to evaluate the planning, problem-solving, logical/arithmetic reasoning, retrieval, and contextual understanding abilities of web agents. Our empirical studies across state-of-the-art LLMs and vision-language models (VLMs), as well as human workers, reveal several challenges for such models to serve as useful assistants in the workplace. In addition to the benchmark, we provide a mechanism to effortlessly generate thousands of ground-truth observation/action traces, which can be used for fine-tuning existing models. Overall, we expect this work to serve as a useful resource to help the community progress toward capable autonomous agents. The benchmark can be found at https://github.com/ServiceNow/WorkArena.

## 1 Introduction

In 2024, the average adult spends about 400 minutes every day interacting with computer software and the internet [DataReportal, 2024]. While some of this time is devoted to productive work, entertainment or creative endeavors, a significant portion is consumed by monotonous, low-value tasks, particularly in enterprise settings. Employees often engage in tedious tasks such as searching for information hidden deeply in knowledge bases, coordinating group discussions to schedule meetings, or filling expense reports in counter-intuitive user interfaces designed for functionality rather than user-friendliness [Bailey and Konstan, 2006, Norman, 2013]. One easily envisions a future where autonomous assistants—AI *agents*—handle these tasks, allowing humans to focus on more complex, skill-intensive work that generates greater value. Recent advances in the reasoning and planning capacities of large language models (LLMs), with developments such as Chain-of-Thought [Wei et al., 2022a], ReAct [Yao et al., 2023], and Tree-of-Thought [Yao et al., 2024] (see [Huang et al., 2024, Wang et al., 2024b] for a review) suggest that this future might be within reach.

While the development of autonomous agents has long been a major research topic both in academia and industry, the recent advances in LLM capabilities have led to a massive surge of interest for

---

[*]Equal contribution.

[†]Core contributors.

[‡]Equal supervision.

38th Conference on Neural Information Processing Systems (NeurIPS 2024) Track on Datasets and Benchmarks.

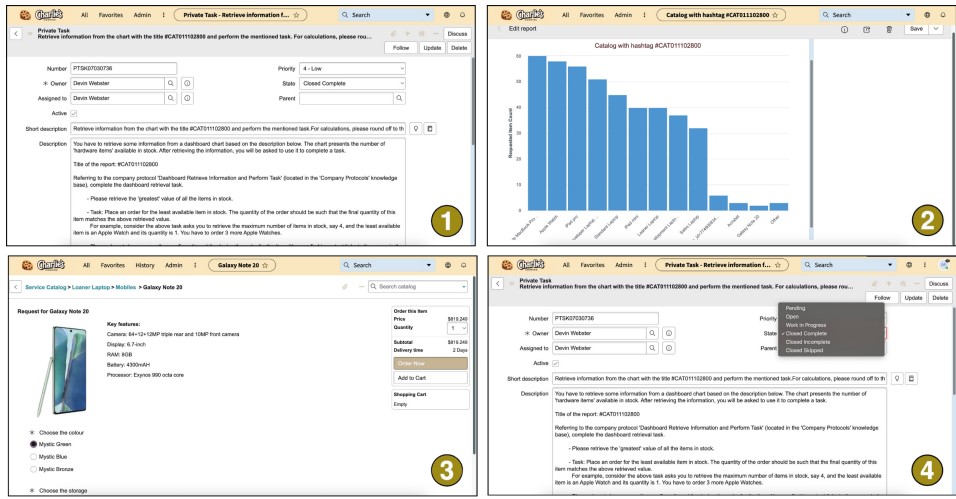

**Figure 1:** Example WorkArena++ task: Restock low inventory items. Here, the agent acts as an IT worker tasked with restocking items that are below some threshold in stock: ① As is common, it receives instructions via a ticket assigned to them in the system; ② it must then read the dashboard to extract all items whose stock count is low; ③ reorder the items from the service catalog to match a minimum stock quantity, and ④ close the ticket assigned to them once the task is completed.

LLM-based autonomous agents [Wang et al., 2024a]. One major application is software control, with a large body of work focused on using LLMs to interact via Application Programming Interfaces (APIs) [Hao et al., 2024, Du et al., 2024]. Another line of research focuses on using LLMs for human-like interactions, by directly manipulating graphical User Interfaces (UIs) on mobile devices [Li et al., 2020, Rawles et al., 2023], desktops [Xie et al., 2024], or websites [He et al., 2024]. This last category encompasses the field of *web agents*, which can automate software interaction even in environments without APIs, improve human productivity, and accessibility for users with disabilities.

In this work we propose *WorkArena++*, a challenging new benchmark to study the proficiency of web agents at solving common knowledge work tasks in enterprise settings. Built on top of the ubiquitous ServiceNow platform, which reported a customer base of more than 7,000 companies and 85% of the Fortune 500 in 2023 [Mastantuono, 2023], it provides a free and robust, yet a realistic evaluation environment for task automation in the workplace. *WorkArena++* expands the *WorkArena* benchmark introduced by Drouin et al. [2024] with 682 challenging new tasks. While the tasks in *WorkArena* are far from being solved by current web agents, they remain predominantly atomic, with simple goals such as filling out a single form with explicit values or navigating explicit menu entries. *WorkArena++* enhances the scale, realism, and complexity of *WorkArena* with composite tasks designed to require skills like problem-solving and memorization, in order to better evaluate the capabilities of web agents at solving complex work tasks. Our contributions are as follows:

- We introduce the WorkArena++ benchmark, considerably expanding the work of Drouin et al. [2024] from 33 to 682 tasks through the addition of realistic office worker trajectories, exemplified in Fig. 1, requiring skills like problem-solving, data-driven decision-making, and more (§ 3). Importantly, this is the first benchmark for web agents to require such complex skills.

- We make a series of technical contributions to WorkArena, such as added visual diversity through the introduction of 10 fictitious companies along with customized UI color schemes, improved database isolation between tasks to facilitate parallel evaluation, a new framework for easily expanding the set of tasks through the composition of low-level building blocks, and the possibility of extracting Oracle-based observation-action traces for fine-tuning (§ 3.3).

- We conduct an empirical study to assess both the difficulty and feasibility of our benchmark, with autonomous agents based on state-of-the-art (visual) language models, both closed and open source, as well as human agents as a baseline. Results indicate that *WorkArena++* presents considerable challenges for current web agents while being reasonably easy for humans (§ 4.3), suggesting its value and relevance as an evaluation benchmark for the scientific community.

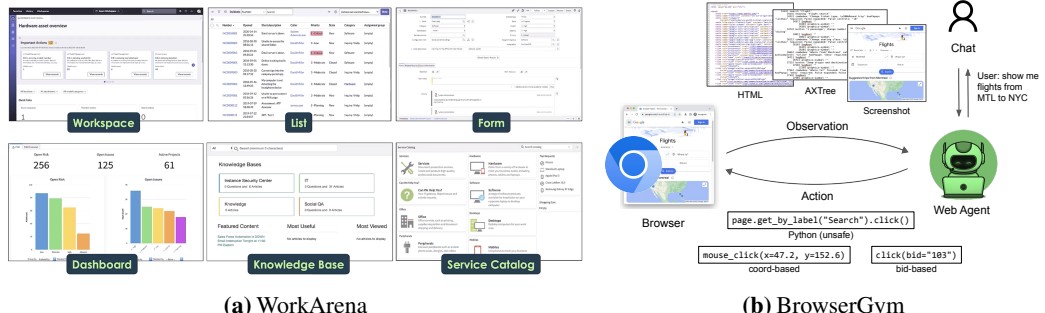

**(a)** WorkArena                  **(b)** BrowserGym

**Figure 2:** Background: (a) In WorkArena, tasks measure the ability of web agents to interact with basic UI components in the ServiceNow platform, illustrated above. (b) In BrowserGym, the agent receives a natural-language goal from a human user via chat. It then perceives the environment (web browser) through a set of multimodal observations (e.g., HTML and screenshot) and controls it via a standardized set of available actions. Reproduced from Drouin et al. [2024] with permission.

# 2 Background

Before diving into our main contribution WorkArena++, we summarize BrowserGym and WorkArena, which respectively provide the environment in which agents interact with the benchmark, and the atomic tasks upon which WorkArena++ is built.

## 2.1 BrowserGym – A Gym Environment for Web Agents

WorkArena++ is integrated into BrowserGym [Drouin et al., 2024] (Fig. 2b), a gym environment that facilitates the design and evaluation of web agents and includes many common benchmarks, such as MiniWob [Shi et al., 2017a, Liu et al., 2018] WebArena [Zhou et al., 2023] and WorkArena [Drouin et al., 2024]. The salient features of BrowserGym include: i) chat-based agent-human interactions, ii) enriched multimodal observations (HTML, accessibility tree [Zhou et al., 2023], screenshot, set-of-marks [He et al., 2024], element coordinates, etc.), and iii) a standardized and flexible action space. In the rest of the paper, all agents interact with WorkArena++ using BrowserGym.

## 2.2 WorkArena

**Figure 3:** In WorkArena(++), the agent interacts with the frontend of a remote-hosted ServiceNow instance via BrowserGym. Task validation then inspects both the state of the database and any open page using backend (REST) and frontend (JS) ServiceNow APIs.

The starting point for our work is WorkArena, the first benchmark to measure the performance of web agents at solving work-related tasks in enterprise settings [Drouin et al., 2024]. Below, we outline some of its key properties.

**Task complexity** While challenging, the tasks included in WorkArena do not require complex problem-solving skills. They rather measure the ability of agents to perform basic interactions with the ServiceNow platform using the main UI components of its user interface, outlined in Fig. 2a. For example, one of the tasks consists in filling out a form after receiving the explicit list of desired values for each field. While solving these tasks is a first step toward achieving anything useful in the workplace, they remain extremely simplistic and trivial for humans.

**Certifiability** An interesting property of WorkArena is that the successful completion of all tasks is certifiable. Each task comes with an *oracle* and a *validator*. The oracle is a human-coded solution that uses browser automation (through Playwright [Microsoft, 2023]) to solve the task. The validator is a function that verifies if the task was solved correctly (e.g., by inspecting the database and the current page) and returns a success reward (0 or 1). Our newly proposed WorkArena++ benchmark builds on the same mechanisms and extends them further to handle more complex tasks.

**Architecture and availability** WorkArena requires agents to interact with remote-hosted clones of the ServiceNow platform called *Personal Developer Instances*. These can be requested for free

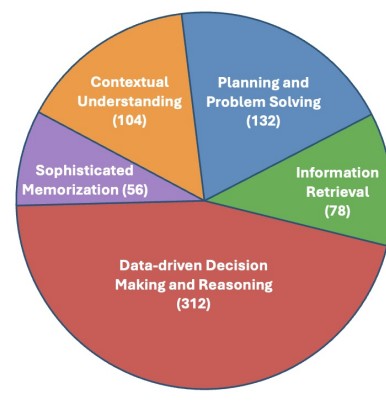

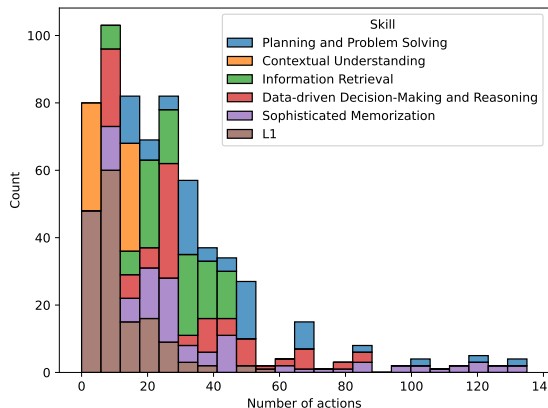

**(a)** Distribution of tasks across skills      **(b)** Number of Oracle actions per task

**Figure 4:** Overview of WorkArena++: a) Distribution of tasks across the skills introduced in §3.2 for all 682 tasks in the L2/L3 sets. b) Task length as estimated by the number of actions required for completion by the Oracle (see §2.2) for all 470 L2/L3 task instances in the *agent curriculum* (§4.1). Tasks from the L1 set are also included for comparison (33 tasks x 5 seeds).

through ServiceNow's Developer Program (see Fig. 3 for an illustration). Of note, WorkArena consists of open-source code that interacts with ServiceNow instance APIs and does not rely on any proprietary code. WorkArena++ follows the same design pattern.

In what remains, we will refer to the tasks from WorkArena as the **L1** tasks (for level 1), while WorkArena++ introduces two new levels **L2** and **L3** with increased difficulty, outlined below.

## 3 WorkArena++: Taking WorkArena to the Next Levels

In WorkArena++, each task consists of a logical combination of simpler atomic tasks, chained together to form a realistic workflow. For example, consider the task of "onboarding a new employee" from the perspective of an IT agent. The process would require the agent to: i) navigate to the appropriate page to create a new user, ii) create a new user account by filling out a form, iii) access a service catalog, iv) order a new laptop, and v) complete a form to assign the laptop to the user in the system. Each of these steps corresponds to a WorkArena L1 task[4]. Next, we provide details on how the tasks in WorkArena++ are categorized across two new levels of difficulty: L2 and L3 (§3.1), and five skill categories (§3.2).

### 3.1 Difficulty Levels L2 and L3

WorkArena++ introduces two new levels of difficulty, L2 and L3, which each cover the same set of 341 workflows presented to the agent in different ways, either as explicit or implicit goals using ServiceNow's ticket mechanism. This makes for $2 \times 341 = 682$ tasks in total. While the workflows to execute in L2 and L3 tasks remain the same, the difficulty is increased in L3 due to the less explicit and more realistic way instructions are provided. Examples of L2 and L3 tasks are showcased in § C.

**L2 – Explicit**      The goal is provided to the agent as detailed instructions sent as a user message in the chat. This message contains the precise steps required to complete the task (e.g., start by navigating to the "users" list, then create a new entry with the following values [...] for an 'onboard user' task.). In addition to following these steps, succeeding at L2 tasks requires thinking, reasoning and contextual understanding.

**L3 – Via ticket + knowledge base**      The goal is provided to the agent in a manner that mimics how human agents receive work assignments as knowledge workers - through a ticket assigned to them (Fig. 1). The ticket includes key details necessary to complete the task (e.g., the name of the new employee for an 'onboard user' task) but does not specify the steps required to solve it. Instead, the agent solving the task is informed that additional instructions can be found in the company's knowledge

---

[4]WorkArena++ mostly re-uses the tasks from L1 as atomic building blocks, but also implements new atomic tasks not in the original L1, such as altering an existing database record.

**Table 1:** Comparing existing web-agent evaluation benchmarks with WorkArena++.

| Benchmark | MiniWoB | WebArena | WorkArena (L1) | WorkArena++ (Ours) |
|---|---|---|---|---|
| **# tasks** | 125 | 190 | 33 | 682 |
| **Env** | Custom web interfaces | Diverse Websites | ServiceNow enterprise platform | ServiceNow enterprise platform |
| **Nature of tasks** | Toy tasks like clicking buttons, filling fields, etc | Real-world inspired tasks from website interactions such as e-commerce and social forums | Basic enterprise software specific tasks such as sorting a list, filling a form | Complex real-world enterprise software tasks performed everyday by knowledge workers |
| **Abilities required** | Basic task and observation space understanding | Complex instruction parsing, advanced UI understanding, information extraction and retrieval | Interface and action adaptability for enterprise software | Planning, logical and arithmetic reasoning, long context understanding and memorization, information extraction and retrieval |
| **Backend** | Selenium | Requires setting up docker for different categories of tasks | Out-of-the-box with browsergym | Out-of-the-box with browsergym |

base if needed. This level is more challenging, as the agent must memorize the task details from a web page and retrieve instructions from a knowledge base before organizing its steps to succeed, requiring consolidating information across multiple sources. We present a comparison of WorkArena++ across different dimensions with various benchmarks in Tab. 1.

To concretely distinguish between an L2 and L3 task, we consider the easy expense management task for example. The goal in L2 is displayed below. While the goal highlights all the necessary steps to complete the task, it does not describe how to accomplish them (e.g. how to use the menu to navigate):

---

**Managing Your Existing Expenses (L2)**

Concretely, you need to complete the following steps:
1. Navigate to the "Expense Lines" module of the "Cost" application.
2. Create a filter for the list to extract all entries where:
    - "Short description" contains "#SERIES-0cbf9a92-4"
3. Delete expense lines with duplicated short descriptions, keeping only one.

---

For the same task, the L3 goal provided to the agent is simply:

---

**Managing Your Existing Expenses (L3)**

Please complete the following task.

---

, accompanied by a ticket that says to refer to a knowledge base article containing the rules for expense management (refer to Fig. 17b). Therefore L3 requires the agent to understand it needs to solve the task from the starting web page, navigate to the KB article, remember the expense management rules and apply them, while in L2 the agent directly has the rule to apply.

## 3.2 Skills and Abilities Evaluated in WorkArena++

Further, all 682 tasks in WorkArena++ fall into one of 5 categories of skills, based on the abilities they require for being solved. The distribution of tasks across categories is displayed in Fig. 4a, and further detailed in § C. We provide a description of each skill category below, along with their number of tasks.

**Planning and problem solving ($66 \times 2$ tasks)** We evaluate these abilities through tasks that require decision-making under constraints to achieve an expected outcome. One notable example consists of scheduling a series of work items within a given time frame while satisfying various constraints based on criticality, duration, and overlap with other items. Other examples include tasks commonly

performed by managers, such as redistributing work among employees based on occupancy, and dispatching work to employees based on expertise.

**Information retrieval (39 $\times$ 2 tasks)** We formulate a series of tasks that require retrieving information from either dashboards or lists (see Fig. 2a) before performing follow-up tasks based on the retrieved values. For example, one task consists of reading a dashboard to find which item is the lowest in stock and restocking by ordering additional items.

**Data-driven decision making and reasoning (156 $\times$ 2 tasks)** This skill, essential to several knowledge work roles, is evaluated through tasks that require interpreting data, performing logical or mathematical reasoning, and taking subsequent actions. One notable example is a task where the agent must select investments to maximize expected return within a limited budget, effectively solving a small instance of a knapsack[5] problem.

**Sophisticated memorization (28 $\times$ 2 tasks)** An essential skill for web agents is the ability to gather information by navigating through a series of pages, memorizing key details along the way, and finally using it to achieve a specific goal. For instance, in the "onboard new employee" task described earlier, the agent receives all the information about the employee, including hardware requirements, and must navigate through multiple pages, filling relevant information at each step to complete the task.

**Contextual understanding through infeasible tasks (52 $\times$ 2 tasks)** Finally, we introduce a set of infeasible tasks to verify if the agent can identify them. We consider two kinds: one where the agent is required to simply declare the task infeasible and another where it must additionally justify its decision. For example, if a task were infeasible due to requesting to fill an inexistent form field, the agent would be evaluated based on its ability to name that field in the justification.

### 3.3 Salient Features of WorkArena++

WorkArena++ does not only augment WorkArena with new tasks, but also includes a number of technical improvements over it which we outline below. For more detail on this, refer to § E.

**Increased visual diversity and realism** WorkArena lacked visual diversity, as all the pages presented to the agent had a similar style. To better assess agents' ability to generalize across different enterprise settings, we introduce 10 fictitious brands, each with distinct styles of the ServiceNow interface. For instance, the "Charlie's Cookies" brand is shown in Fig. 1. In WorkArena++, a company brand is randomly sampled at the start of each task, enhancing realism and visual diversity. This changes the colors of visual elements as well as the company logo. More visual diversity could be addressed in future works.

**Standardization and task isolation** For a benchmark to be robust, it must ensure a standardized level of difficulty, regardless of variables like parallel inference or the hardware used for experiments. Achieving this on a remote-hosted instance without access to proprietary code presents a challenge. In WorkArena++, we enhance robustness through several measures. First, we provide an installer that configures the instance with standardized system parameters, UI themes, knowledge bases, and layouts for components like lists and forms. Second, we implement sandboxing by running each task in a new user account, created at its start and automatically deleted at its end. This allows agents to interact freely with the system (e.g., changing visible columns in a list) without affecting subsequent tasks. Together, these improvements result in a more robust benchmark.

**Extendability and extraction of fine-tuning data** Tasks in WorkArena++ are created by carefully composing the *oracle* and *validator* functions (see §2.2) of simpler tasks, such as those in the L1 set. Notably, our framework allows the combination of simple oracle functions to generate human-coded ground truths for more complex compositional tasks. Additionally, this framework facilitates the collection of observation-action traces, regardless of the task's length and complexity, providing valuable fine-tuning data for LLMs and VLMs in web-agent interactions.

## 4 Experiments

We now present a series of experimental results on WorkArena++. As will be shown, our proposed benchmark poses a significant challenge for state-of-the-art web agents while being relatively simple

---

[5]https://en.wikipedia.org/wiki/Knapsack_problem.

for humans to solve. This contrast underscores the benchmark's potential to drive advancements in the field, providing the community with a valuable tool for evaluating and improving web agents.

## 4.1 Evaluation Curriculum: Standardizing WorkArena++ as a Benchmark

Each WorkArena++ task can be instantiated with variability from thousands of valid configurations per task. Hence, the benchmark can be viewed as a rich distribution over task instances. In order to make the cost of evaluation accessible, improve reproducibility, and have a uniform test of various skills required for solving the tasks, we propose a standardized way to sample a collection of task instances, which we refer to as a curriculum. Concretely, we provide a mechanism that takes a numerical seed as input and can produce an arbitrary number of task instances, sampled uniformly across skills in a reproducible way. We reserve seeds 0-9 for evaluation and the remaining seeds may be used for agent tuning[6]. Additional details in § D.

## 4.2 Agent Design

We mostly follow the same agent design as Drouin et al. [2024], which consists in using an LLM augmented with chain-of-though prompting [Wei et al., 2022b] to produce the next best action for solving the task, based on the current observation of the web browser.

**Observation space**   The main elements presented to our web agents are the current goal, the current page's accessibility tree (AXTree) [Zhou et al., 2023] which can be seen as a compressed representation of the HTML, and the error message (if any) that resulted from the execution of the previous action. Additionally, our VLM-based agent is given a screenshot of the current page augmented with set-of-marks [He et al., 2024]. As most tasks in WorkArena++ require long context understanding, we also add to the prompt the history of their previous actions and thoughts (from chain of thoughts) since the start of the episode. This simple mechanism provides a crude memorization mechanism to otherwise memory-less agents, giving them more chances of solving L2 and L3 tasks.

**Action space**   All our agents use the high-level action space from BrowserGym, restricted to the `chat`, `infeas` and `bid` action sets, which allow sending messages to the chat, declaring a task infeasible, and interacting with the current webpage via primitives using element identifiers (`bid` attribute), e.g., clicking on an element, filling a text box, etc. Our agents are restricted to producing only one action at a time (as in [Drouin et al., 2024]), and implement a retry mechanism that can re-prompt the agent up to 4 times in case of parsing errors in their output.

**Language models**   We evaluate closed-source models GPT-3.5 and GPT-4o [OpenAI, 2023] and study the impact of providing screenshots of the pages with GPT-4o vision. On the open-source side, we evaluate Llama3-70b [Meta, 2024] and Mixtral-8x22b [Jiang et al., 2024], deployed using Hugging Face's Text Generation Inference (TGI) library on 4 A100 GPUs. We use a maximum context length of 40K tokens for GPT-4o, 15K for GPT-3.5, 8K for Llama3 and 32K for Mixtral. To ensure that prompts do not exceed those limits, we progressively truncate the accessibility tree from the end until it fits in the context. For more information on agent design and the setup, please refer to § B.

**Maximum number of steps**   For budget reasons, we run our agents for a maximum of 50 time-steps before the tasks are terminated. According to our oracle analysis in Fig. 4b, this gives our agents the chance to solve most of the tasks in our *evaluation curriculum*.

## 4.3 Agent Results

We evaluate all baseline agents on the standardized curriculum introduced in §4.1 and report the results in Tab. 2. A notable takeaway is that all agents, whether closed-source or open-source, and regardless of being LLM or VLM-based, generally fail to achieve any reasonable success on WorkArena++, despite performing reasonably well on existing benchmarks. Only GPT-4o and GPT-4o-v succeed at some tasks, particularly memorization tasks within the L2 set, with no successes observed in the L3 set. Interestingly, we observe that, in contrast to its unimodal counterpart GPT-4o, the GPT-4o-v agent succeeds at solving a few retrieval tasks involving reading values off charts, suggesting that the

---

[6]To reduce standard error, users can average across multiple seeds between 0-9. Also, it is encouraged to tune the agent using different benchmarks, e.g. WebArena, to better evaluate generalization, but if one must use WorkArena++ for tuning, we encourage to avoid seeds 0-9.

**Table 2:** Success rate$_{\pm\text{Standard error}}$ (SR $_{\pm\text{SE}}$) of all agents on MiniWoB, WorkArena, and WebArena, with numbers reported in %. Bolded numbers represent the average success rate over the entire corresponding benchmark. Results on WorkArena L1, WebArena and MiniWoB are extracted from Drouin et al. [2024]. Human evaluation numbers for MiniWoB and WebArena are taken from Humphreys et al. [2022] and Zhou et al. [2023] respectively. The number of task instances is for the agent curriculum. For more detail on human curriculum, refer to § A.

| Task Category (task instances count) | Agent Curriculum (full benchmark) | | | | | Human Curriculum (subset of tasks) | |
|---|---|---|---|---|---|---|---|
| | GPT-3.5 | GPT-4o | GPT-4o-v | Llama3 | Mixtral | Human | GPT-4o |
| **WorkArena L3** (235) | **0.0** $_{\pm0.0}$ | **0.0** $_{\pm0.0}$ | **0.0** $_{\pm0.0}$ | **0.0** $_{\pm0.0}$ | **0.0** $_{\pm0.0}$ | **93.9** $_{\pm3.4}$ | **0.0** $_{\pm0.0}$ |
| Contextual Understanding (32) | 0.0 $_{\pm0.0}$ | 0.0 $_{\pm0.0}$ | 0.0 $_{\pm0.0}$ | 0.0 $_{\pm0.0}$ | 0.0 $_{\pm0.0}$ | 87.5 $_{\pm11.7}$ | 0.0 $_{\pm0.0}$ |
| Data-driven Decision-Making (55) | 0.0 $_{\pm0.0}$ | 0.0 $_{\pm0.0}$ | 0.0 $_{\pm0.0}$ | 0.0 $_{\pm0.0}$ | 0.0 $_{\pm0.0}$ | 100.0 $_{\pm0.0}$ | 0.0 $_{\pm0.0}$ |
| Planning and Problem Solving (44) | 0.0 $_{\pm0.0}$ | 0.0 $_{\pm0.0}$ | 0.0 $_{\pm0.0}$ | 0.0 $_{\pm0.0}$ | 0.0 $_{\pm0.0}$ | 87.5 $_{\pm11.7}$ | 0.0 $_{\pm0.0}$ |
| Information Retrieval (56) | 0.0 $_{\pm0.0}$ | 0.0 $_{\pm0.0}$ | 0.0 $_{\pm0.0}$ | 0.0 $_{\pm0.0}$ | 0.0 $_{\pm0.0}$ | 100.0 $_{\pm0.0}$ | 0.0 $_{\pm0.0}$ |
| Sophisticated Memorization (48) | 0.0 $_{\pm0.0}$ | 0.0 $_{\pm0.0}$ | 0.0 $_{\pm0.0}$ | 0.0 $_{\pm0.0}$ | 0.0 $_{\pm0.0}$ | 91.7 $_{\pm8.0}$ | 0.0 $_{\pm0.0}$ |
| **WorkArena L2** (235) | **0.0** $_{\pm0.0}$ | **3.0** $_{\pm1.1}$ | **3.8** $_{\pm1.3}$ | **0.0** $_{\pm0.0}$ | **0.0** $_{\pm0.0}$ | **93.9** $_{\pm3.4}$ | **2.1** $_{\pm2.0}$ |
| Contextual Understanding (32) | 0.0 $_{\pm0.0}$ | 0.0 $_{\pm0.0}$ | 0.0 $_{\pm0.0}$ | 0.0 $_{\pm0.0}$ | 0.0 $_{\pm0.0}$ | 100.0 $_{\pm0.0}$ | 0.0 $_{\pm0.0}$ |
| Data-driven Decision-Making (55) | 0.0 $_{\pm0.0}$ | 0.0 $_{\pm0.0}$ | 0.0 $_{\pm0.0}$ | 0.0 $_{\pm0.0}$ | 0.0 $_{\pm0.0}$ | 84.6 $_{\pm10.0}$ | 0.0 $_{\pm0.0}$ |
| Planning and Problem Solving (44) | 0.0 $_{\pm0.0}$ | 0.0 $_{\pm0.0}$ | 0.0 $_{\pm0.0}$ | 0.0 $_{\pm0.0}$ | 0.0 $_{\pm0.0}$ | 100.0 $_{\pm0.0}$ | 0.0 $_{\pm0.0}$ |
| Information Retrieval (56) | 0.0 $_{\pm0.0}$ | 0.0 $_{\pm0.0}$ | 3.6 $_{\pm2.5}$ | 0.0 $_{\pm0.0}$ | 0.0 $_{\pm0.0}$ | 100.0 $_{\pm0.0}$ | 0.0 $_{\pm0.0}$ |
| Sophisticated Memorization (48) | 0.0 $_{\pm0.0}$ | 14.6 $_{\pm5.1}$ | 14.6 $_{\pm5.1}$ | 0.0 $_{\pm0.0}$ | 0.0 $_{\pm0.0}$ | 91.7 $_{\pm8.0}$ | 8.3 $_{\pm8.0}$ |
| **WorkArena L1** (33 × 10 seeds) | **6.1** $_{\pm1.3}$ | **42.7** $_{\pm2.7}$ | **41.8** $_{\pm2.7}$ | **17.9** $_{\pm2.1}$ | **12.4** $_{\pm1.8}$ | – | – |
| **MiniWoB** (125 × 5 seeds) | **43.4** $_{\pm1.6}$ | **71.3** $_{\pm1.5}$ | **72.5** $_{\pm1.5}$ | **68.2** $_{\pm1.2}$ | **62.4** $_{\pm1.6}$ | *93.5* | – |
| **WebArena** (812) | **6.7** $_{\pm0.9}$ | **23.5** $_{\pm1.5}$ | **24.0** $_{\pm1.5}$ | **11.0** $_{\pm1.1}$ | **12.6** $_{\pm0.5}$ | *78.2* | – |

vision modality can be beneficial in WorkArena++. Additionally, as expected, the GPT-4o-based agent significantly outperforms its GPT-3.5 counterpart.

These results raise important questions: i) Are these tasks actually solvable? and ii) Why do the agents fail? In what follows, we address each of these questions through human evaluation (§ 4.4) and error analysis (§ 4.5), respectively.

## 4.4 Human Evaluation

To assess the feasibility of the benchmark and measure the gap between humans and agents, we conducted a study with 15 human subjects tasked with solving WorkArena++ tasks. Given the limited number and availability of subjects, we devised a shorter curriculum comprising 98 task instances, sampled uniformly across skills and the L2/L3 sets. Each participant solved a subset of the tasks using a custom-made evaluation tool that exactly matched the interface available to agents. We report these results in Tab. 2 under the Human Curriculum column, along with the corresponding performance of our GPT-4o agent on the same subset of tasks. The numbers are striking, with an overall 93.9% success rate for humans and 2.1% for GPT-4o. These establish WorkArena++ as a valuable benchmark that is both solvable and relatively straightforward for humans, while being challenging for state-of-the-art LLMs, emphasizing its value as a new milestone for the community.

We note that all subjects consented to participate in the study without compensation. Most had little to no familiarity with ServiceNow products and underwent only a brief training session, consisting of a 15-minute video outlining the components in Fig. 2a, followed by 15 minutes of self-guided exploration on the platform. Details on the task curriculum, training received, demographics, and the evaluation platform are included in § A.

## 4.5 Error Analysis

To understand the poor performance of agents on WorkArena++, we conducted an in-depth study of their execution traces. We focused on the best-performing open- and closed-source models, Llama3 and GPT-4o. This analysis identified salient types of errors, which sheds light on current model limitations and highlight areas for improvement.

**Information Retrieval** The models tend to successfully navigate to the correct information sources (e.g., dashboards). However, they occasionally fail to accurately retrieve the relevant information from the observations. Furthermore, agents sometimes fail to identify relevant elements on the page and attempt to act on incorrect ones.

**Exploration** Some tasks require to explore the page for hidden information, such as opening different tabs in a form or expanding a section of foldable elements. The agents often struggle due to a lack of curiosity, leaving them stuck in their location.

**Hallucination** *Made-Up Actions:* The models sometimes hallucinate actions that would be convenient for the task at hand, but that are not available in BrowserGym. *Imaginary Buttons:* Similarly, we observed cases of interaction with made-up buttons that would solve tasks in one click, such as buttons that create the exact filter required. *Asking for Help:* When confused about the next steps, models sometimes ask for help via chat, indicating a lack of confidence or capacity in planning the next steps.

**Goal Understanding** *Thought/Action Consistency:* There are instances where the models' thought process correctly identifies the next action, but the action produced is different and incorrect. This inconsistency undermines performance. *Low-Level Understanding in L3 Tasks:* In more complex L3 tasks, the models fail to comprehend the necessary subtasks fully. For example, they might start by attempting to modify ticket values that are locked, showing a misunderstanding of the task requirements.

**Action Consequences Assessment** *Hallucinated Consequences:* The models often hallucinate the consequences of their actions, believing they have made progress on the task when no actual progress has occurred. *Repeated Actions:* When an action does not change the state of the webpage, models tend to retry the same action repeatedly instead of trying a different approach.

These errors illustrate the current limitations of state-of-the-art web agents in handling complex enterprise tasks. Addressing these issues is crucial for developing more reliable and effective autonomous agents capable of performing real-world knowledge work. Detailed examples are included in § F.

## 5 Related Work

Early benchmarks for web agents primarily utilized synthetic web environments where agents executed low-level keyboard and mouse actions [Shi et al., 2017b,a, Liu et al., 2018]. More recently, Zhou et al. [2023] introduced WebArena, comprising 190 tasks based on realistic websites that emulate real-world domains such as e-commerce, social forums, and content management. OSworld [Xie et al., 2024] introduces a scalable, real computer environment for benchmarking multimodal agents, supporting task setup, execution-based evaluation, and interactive learning across operating systems.

In terms of datasets, Deng et al. [2023] proposed Mind2Web, a large-scale collection of 2,000 web interactions from 137 websites curated by human annotators. Similarly, Lù et al. [2024] introduced WebLINX, a curated dataset of web interactions with 2337 expert demonstrations from 155 different real-world websites. He et al. [2024] proposed 300 information-retrieval tasks from 15 real-world consumer websites, evaluating WebVoyager, a vision-based web agent's capabilities. WorkArena [Drouin et al., 2024] focuses on real-world enterprise software applications by including 33 interactions tasks representative of realistic workflows typically performed by knowledge workers.

Building on this foundation, WorkArena++ introduces tasks requiring advanced skills like problem-solving and data-driven decision-making. Unlike previous benchmarks, WorkArena++ evaluates agents on their ability to perform complex, multi-step tasks that closely mimic real-world enterprise scenarios. Additionally, WorkArena++ emphasizes task isolation, uniform setup, and robust evaluation guidelines, enabling fair comparisons and reproducibility. This approach makes WorkArena++ unique in evaluating the capabilities of LLM and VLM-based web agents.

## 6 Conclusion and Future Work

We propose WorkArena++, a novel benchmark consisting of 682 tasks to evaluate web agents by mimicking realistic workflows performed routinely by knowledge workers using enterprise software. WorkArena++ tests various complex skills of LLM and VLM-based agents including planning, decision-making, retrieval, logical and arithmetic reasoning, as well as the ability to identify infeasible tasks. Our benchmark promotes standardized evaluation, realistic visual diversity, and provides a method for generating large amounts of fine-tuning data in the form of web-interaction traces. Empirical evaluations reveal that state-of-the-art LLMs and VLMs struggle with our benchmark, while humans achieve extremely high performance. Through our error analyses and qualitative studies, we hope WorkArena++ will be a significant step towards developing more capable autonomous web agents.

In future work, we aim to continue developing new sets of tasks on the ServiceNow platform. Of particular importance would be tasks for evaluating safety and cybersecurity around agents as well as a hidden test set for hosting competitions. Furthermore, our framework offers the ability to generate a vast amount of fine-tuning data through web interaction traces to train more robust LLM and VLM-based web agents. Our ultimate goal is to close the significant gap between autonomous agents and humans on WorkArena++ and other benchmarks.

## 7 Limitations and Potential Societal Impacts

**Limitations**  While WorkArena++ includes diverse and realistic workflows, it does not exhaustively cover all possible knowledge-work tasks and personas. Achieving comprehensive coverage would require thousands of additional tasks. However, our task set is designed to be easily extendable by the community, and we welcome such contributions. Additionally, while this work evaluates the reasoning abilities of LLM-based web agents, it does not assess their safety and robustness against various malicious behaviors, which remains an important barrier to their adoption in real-world settings. Moreover, the benchmark does not include tasks that require interaction with software external to the ServiceNow platform, which would improve diversity and realism. We leave such assessments to OS-level benchmarks like that of Xie et al. [2024]. Finally, we note that additional open and closed-source LLMs and VLMs could have been included in the experiments, particularly those with extremely long context lengths, such as Gemini 1.5 pro with 1 million tokens [Reid et al., 2024].

**Societal Impacts**  This work is likely to inspire the development of agents as valuable workplace assistants, positively impacting society in several ways. It can increase productivity, enabling agents to handle more complex and value-creating tasks. Additionally, it can improve accessibility for impaired users, potentially opening new job opportunities. However, there are potential negative impacts. Advanced agents may lead to job displacement as such systems take over human tasks. Reliance on these agents raises data privacy and security concerns and could erode human skills and decision-making abilities over time. Moreover, the significant computational resources required to support these agents lead to substantial energy use, contributing to negative environmental impacts.

## Acknowledgments and Disclosure of Funding

The authors are grateful to the individuals who participated in our evaluation study: Arjun Ashok, Aayush Bajaj, Ghazwa Darwiche, Jerry Huang, Raymond Li, Marie-Ève Marchand, Étienne Marcotte, Shravan Nayak, Sébastien Paquet, Soham Parikh, Fanny Rancourt, Gopeshh Subbaraj, Jordan Prince Tremblay, and Andrew Williams. Your contributions were invaluable in showcasing that, for now, humans are still the reigning champions of intelligence. We also extend our thanks to Chris Pal, Issam Laradji, and David Vazquez for their insightful feedback and suggestions, which were almost as brilliant as our participants' defense of human ingenuity.

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

# Appendix

## Table of Contents

## Contents

## A  Human Evaluation – Additional Details

This section provides additional details on our human evaluation study, where humans were tasked with solving WorkArena++ tasks.

### A.1  Participants

We recruited a cohort of 15 volunteers with varying levels of familiarity with ServiceNow products. This group included core members of ServiceNow's research team and students from local institutions, namely the University of Montreal and the *École de technologie supérieure*. All participants provided informed consent as outlined in the consent form (see Fig. 5). The demographic profile of the participants is depicted in Fig. 6 (see § A.5 for a discussion of the representativeness of the cohort).

## A.2 Protocol

An in-person event was conducted at the ServiceNow Montreal office. The session began with a brief training, during which participants watched a 15-minute recorded talk (slides shown in Fig. 7). After the talk, participants engaged in 15 minutes of hands-on self-exploration of ServiceNow products using the same free Personal Developer Instances employed in the main study. Following this self-exploration period, each participant was asked to solve up to seven tasks individually on the *evaluation platform* described below. The curriculum from which these tasks were sampled is also described below. Participants were instructed not to discuss the tasks among themselves. They were informed that both their time to resolution and success rate would be recorded. The exact web page that participants used for guidelines on the day of the event is available in the benchmark's GitHub Wiki.

## A.3 Task curriculum

Due to the limited time and availability of human evaluators, we had to limit the size of the curriculum used in the study. Hence, we evaluated humans based on a curriculum of 98 task instances, sampled uniformly at random across skills (§ 3.2), and considered both their L2/L3 variants (49 task instances x 2 levels). The exact curriculum used can be retrieved from the benchmark's codebase.

## A.4 Evaluation platform

The human evaluators were asked to solve tasks in a UI that identically matches that available to the agents (i.e., BrowserGym [Drouin et al., 2024]), except for the addition of a "Human Evaluation Console" overlay (see Fig. 8). This console provided functionalities such as a "Validate" button that allowed them to check if they had completed the task successfully and a "Give up" button that allowed them to abandon the task (counting as a failure). Other noteworthy features include: i) an auto-validation mechanism that constantly checked the completion status of the page in real time, complementing the validation button, and ii) the console's movability to prevent blocking the evaluator's view of the page. Finally, note that, humans did not receive feedback while completing the tasks; they could only see if the task was currently incomplete or solved. The code for the human evaluation platform is included in our main codebase, along with a command-line program to launch it.

## A.5 Limitations

**Learning effect:** We noticed that participants became increasingly efficient at solving tasks as they progressed through their assigned curriculum. We hypothesize that this was primarily due to gaining familiarity with the product's user interface. While our protocol does not directly account for this learning effect, we note that it is unlikely that the evaluators learned to solve any given task since most were not asked to perform the same task twice. Among our 15 evaluators, only 3 had curricula that included a repeated task, and, importantly, these tasks were always presented under another level (L2/L3) with a different seed. The significant difference between the presentation of goals and instructions in L2/L3 task acts as further mitigation.

**General announcements to participants:** During the evaluation, multiple participants encountered the same issue due to the user interface being counterintuitive, which led to a general announcement.

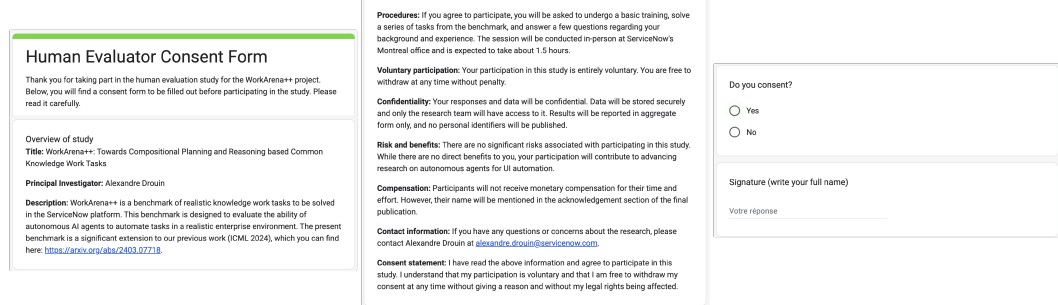

**Figure 5:** Consent form signed by all human evaluators prior to participating in the study.

The announcement clarified that a ticket would not be marked as "Closed - Complete" until the form was saved via the "Update" button. This is important for L3 tasks, as such tasks are only considered complete when the agent marks the ticket as "Closed - Complete." Although most participants correctly changed the ticket's state field, they did not save their changes, resulting in the task being considered incomplete. After our announcement, this error did not recur. It is important to note that this hint cannot account for the significant performance difference between humans and the LLM-based agents, as the LLMs fail to complete the initial steps of the L3 tasks, never approaching this point in the task's trajectory.

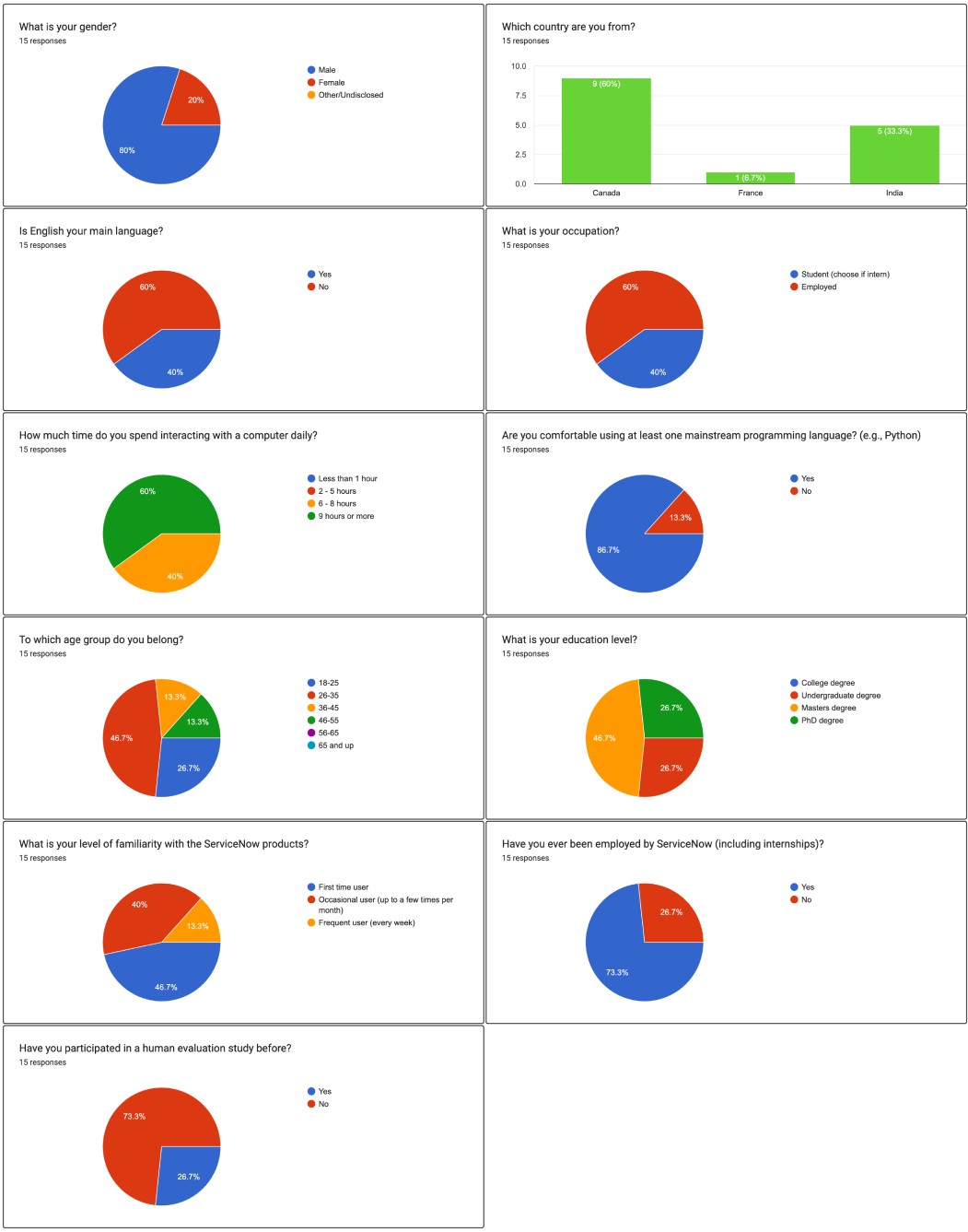

**Figure 6:** Demographic profile of study participants. Each subfigure presents the distribution of responses to a specific question, including the available answer choices and the corresponding number of responses.

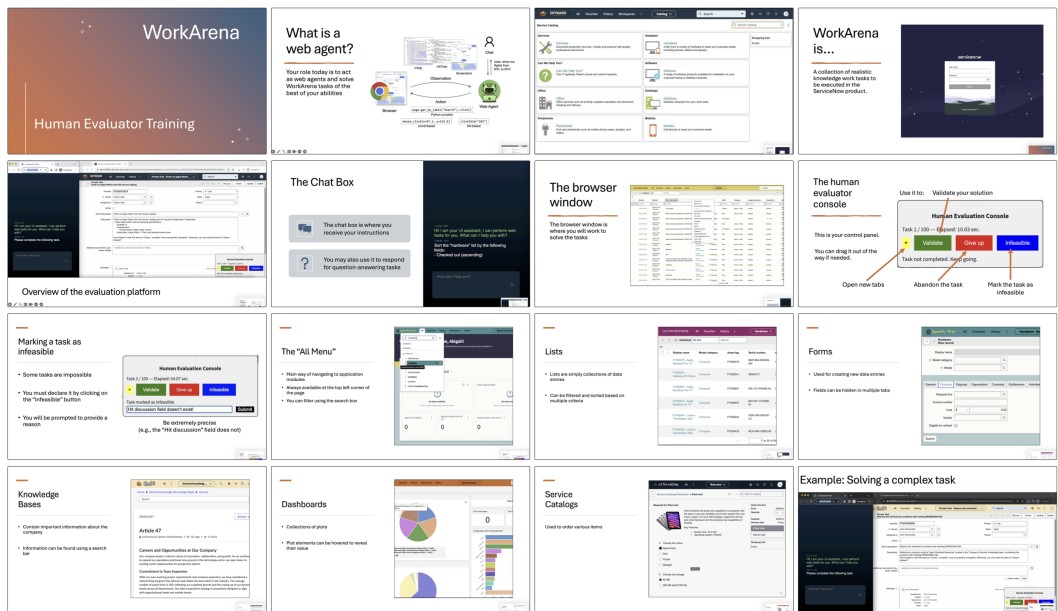

**Figure 7:** Slides for the 15-minute presentation used to train human evaluators. To be viewed from left to right.

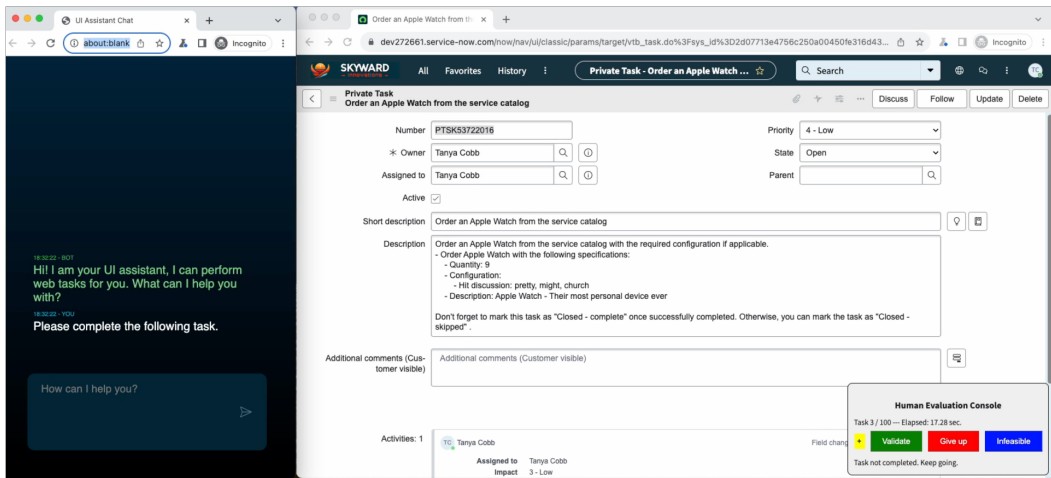

**Figure 8:** Human evaluation platform. The interface consists of the BrowserGym chat (left) and browser (right) windows, along with the "Human Evaluation Console" (bottom-right) as an overlay.

**Cohort representativeness:**    The demographics data reveals that approximately 75% of human evaluators have worked at ServiceNow, all of them with undergrad degrees, and half with advanced degrees. It is likely that this is not representative of the general user demographics of ServiceNow. Hence, it is normal to wonder if the high performance of human evaluators accurately characterizes the general cognitive complexity of the tasks for arbitrary first-time users. We discuss multiple aspects:

- **ServiceNow Employees:** Out of our evaluators, 11/15 have been employed by ServiceNow. However, this number must not be interpreted as "people who have interacted with the product before". In fact, many of these people have roles that do not involve interaction with the product. For example, 4 of these 11 evaluators were student researchers who, by the nature of their work, are not exposed to the product. Moreover, we emphasize that 46.7% of our evaluators declared being first-time users of the product, and 86.7% claim to use the product at most a few times per month. Finally, we stress that the tasks in WorkArena++ mostly evaluate

planning and reasoning skills, beyond basic interaction with the product that some evaluators could be familiar with.

- **Study degrees:** This bias is present in our pool of evaluators. Nevertheless, we believe it does not invalidate our conclusions, since none of our tasks involve skills that one would acquire through an advanced degree. Among the "general user demographics" of ServiceNow, all users must be able to manipulate basic UI components, such as forms, lists, etc. Beyond that, succeeding at our benchmark involves following a clear set of instructions and basic reasoning that should be well within reach for anyone with the ability to interact with the software. Hence, while this bias is present, we argue that it is not a significant driver of success on the benchmark.

- **Zero-shot models:** We stress that agents based on LLMs cannot be characterized as "first-time users" of ServiceNow. Our exploration revealed that these models have extensive knowledge of the ServiceNow platform, such as very detailed knowledge of available APIs, detailed information about the underlying data structure, and knowledge of how to solve certain tasks in the product. This likely stems from training on publicly available documentation and discussions in public support forums. Hence, we believe that the relative lack of exposure to the platform of most human evaluators and the knowledge held by LLMs/VLMs helps in normalizing possible discrepancies in our human evaluation.

**Could training and self-exploration explain human performance?**    The training given to human evaluators (§ A.2) contains very high-level information about which UI components are included in tasks and how to use our human evaluation platform (see Fig. 7). This in itself does not reveal information that helps to solve tasks. In the 15 minutes of self-exploration, the humans learn how to manipulate the UI components, a skill that the agents might indeed acquire through fine-tuning. However, as emphasized by our error analysis (§ 4.5), most failures of agents are due to mistakes in planning and reasoning, not basic UI manipulation. Furthermore, as previously explained, LLMs/VLMs have access to a wealth of information on ServiceNow products, potentially giving agents an advantage well beyond knowledge of basic UI manipulation.

**Is there a maximum number of actions for human evaluators?**    To simplify the evaluation process, we did not impose an explicit limit on the number of actions that human evaluators could take. In contrast, AI agents are subject to an action limit, not to impose additional constraints, but to account for practical resource limitations, such as credit usage and rate limits. The number of actions allowed for AI agents is designed to be sufficient for task completion.

# B   Agent Design

Below are the general design choices of our LLM/VLM-based web agent with chain-of-thought prompting [Wei et al., 2022b].

**Language models:**   Our study distinguishes between closed- and open-source LLMs. For the closed-source segment, we evaluate **GPT-3.5** (`gpt-3.5-turbo-1106`, 16K context) and **GPT-4o** [OpenAI, 2023] (`gpt-4o-2024-05-13`, 128K context), through OpenAI's API. We also explore the effect of providing the screenshot of the page using GPT-4o vision and Set-of-Mark [Yang et al., 2023] as proposed in WebVoyager [He et al., 2024]. In the realm of open-source LLMs, we evaluate both **Llama3-70b** [Meta, 2024] (`meta-llama-3-70B-instruct`, 8K context) and **Mixtral 8x22B** [Jiang et al., 2024] (`open-mixtral-8x22b`, 64K context). These model are deployed using Hugging Face's Text Generation Inference (TGI) library on 4 A100 GPUs.

**Observation space:**   Our observation space is composed of the goal, the current page's HTML and/or AXTree,[7] the currently focused element, and the error from the previous action if any. We also augment each element with two extra boolean properties provided by BrowserGym, `clickable` and `visible`. Finally, when using GPT-4o vision we also give the model a screenshot of the current page, augmented with set-of-marks [He et al., 2024] using the element identifiers (`bid`) provided by BrowserGym.

**Action space:**   We use BrowserGym's high-level action space with `chat`, `infeas` and `bid` primitives [Drouin et al., 2024] which respectively allow the agent to send messages to the chat (`send_msg_to_user(text)`, necessary for information retrieval tasks), to declare the task infeasible (`report_infeasible(reason)`, necessary for infeasible tasks[8]), and to interact with the page's HTML elements using their unique identifiers (e.g., `click(bid)`, `type(bid, text)` etc.). The `bid` primitives rely on the unique `bid` attribute given by BrowserGym to each HTML element, which is made available textually in the HTML and AXTree as well as visually through set-of-marks in the screenshots. The full action space is described to the agent in the prompt, with individual examples of valid function calls for each primitive. For an example prompt and the corresponding screenshot with set-of-marks, see Fig. 9 and Fig. 10.

**History:**   To extend the horizon window of our agent, at each time step we re-inject into the agent's prompt the history of all previous actions and thoughts (from chain-of-thought) since the start of the episode. This gives our agent a chance to recall its previous thoughts, thereby providing a crude memorization mechanism to otherwise memory-less agents, giving them more chances (theoretically) of solving memory-intensive L2 and L3 tasks.

**Zero-shot examples:**   In the prompt, we provide a single generic example of how the chain-of-thought and action outputs should be formatted. This contrasts with other methods [Kim et al., 2023] where task-specific few-shot examples are provided, yet aligns with our objective of developing zero-shot agents able to solve a large range of new tasks.

**Parse and retry:**   Once the LLM provides an answer, we have a parsing loop that can re-prompt the agent up to 4 times to make it aware of a parsing mistake. This can save the agent from making basic mistakes and is mainly useful for less capable LLMs. Once parsed, the action is executed via BrowserGym, which moves to the next step.

**Prompt truncation:**   We use a maximum prompt length of 40K tokens for GPT-4o, 15K for GPT-3.5, 8K for Llama3 and 32K for Mixtral. When the prompt is too large, we progressively truncate the HTML and AXTree from the end until it fits the maximum allowed number of tokens.

**Prompt tuning:**   We tuned prompts to get the best performing agents following the methodology of Drouin et al. [2024].

---

[7]On WebArena and WorkArena we only use AXTrees because HTML is prohibitively large. On MiniWoB we use both AXTree and HTML as it consistently gives the best performance.

[8]This new `infeas` primitive was introduced specifically for WorkArena++, and is made compatible with other benchmarks with infeasible tasks such as WebArena.

**Example Prompt - L1 - Order Sales Laptop task**

```
# Instructions
Review the current state of the page and all other information to find the best
possible next action to accomplish your goal. Your answer will be interpreted
and executed by a program, make sure to follow the formatting instructions.

## Goal:
Go to the hardware store and order 6 "Sales Laptop" with configuration
{'Additional software requirements': 'Slack, Zoom, Google Workspace, HubSpot, Adobe Creative Cloud',
'Adobe Acrobat': True, 'Adobe Photoshop': False, 'Microsoft Powerpoint': False, 'Siebel Client': False}

# Observation of current step:

## AXTree:
Note: [bid] is the unique alpha-numeric identifier at the beginning of lines for each element in the
AXTree. Always use bid to refer to elements in your actions.
Note: You can only interact with visible elements. If the "visible" tag is not
present, the element is not visible on the page.

RootWebArea 'Catalog | ServiceNow'
...
    [a] Iframe 'Main Content', visible
      RootWebArea 'Catalog', focused
...
                    [a251] heading 'Hardware', clickable, visible
                      [a252] link 'Hardware', clickable, visible
...
              [a261] link '', clickable, visible
                [a262] table '', visible
                  [a263] rowgroup '', visible
                    [a264] row '', visible
                      [a265] gridcell '', visible
                      [a268] gridcell 'Hardware. Order from a variety of hardware to meet your business
                      needs, including phones, tablets and laptops. Order from a variety of hardware to meet
                      your business needs, including phones, tablets and laptops.', clickable, visible
                        [a269] link 'Hardware. Order from a variety of hardware to meet your business
                        needs, including phones, tablets and laptops.', clickable, visible
                          [a270] heading 'Hardware', visible
...

## Focused element:
bid='a85'

# History of interaction with the task:
...

# Action space:

Note: This action set allows you to interact with your environment. Most of them
are python functions executing playwright code. The primary way of referring to
elements in the page is through bid which are specified in your observations.
13 different types of actions are available.
...
fill(bid: str, value: str)
    Description: Fill out a form field. It focuses the element and triggers an input event with the
    entered text. It works for <input>, <textarea> and [contenteditable] elements.
    Examples:
        fill('237', 'example value')
        fill('45', 'multi-line\nexample')
        fill('a12', 'example with "quotes"')
...
send_msg_to_user(text: str)
    Description: Sends a message to the user.
    Examples:
        send_msg_to_user('Based on the results of my search, the city was built in 1751.')

Only a single action can be provided at once. Example:
fill('a12', 'example with "quotes"')
...
# Concrete Example

Here is a concrete example of how to format your answer.
Make sure to follow the template with proper tags:

<think>
From previous action I tried to set the value of year to "2022",
using select_option, but it doesn't appear to be in the form. It may be a
dynamic dropdown, I will try using click with the bid "a324" and look at the
response from the page.
</think>

<action>
click('a324')
</action>
```

**Figure 9:** Example prompt of our LLM-based agent. Some parts are truncated (...) for clarity.

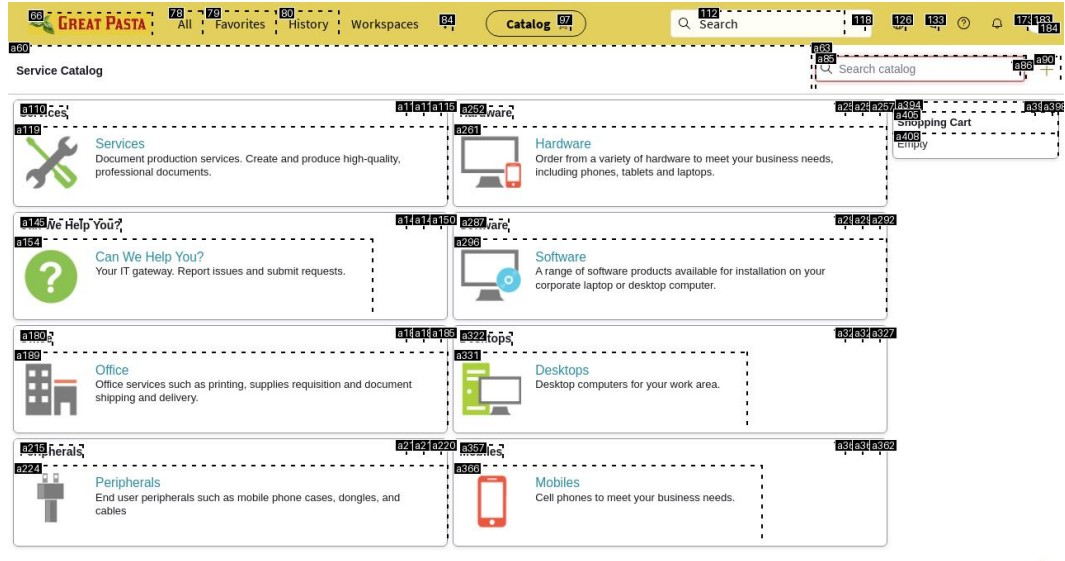

**Figure 10:** Example screenshot with set-of-marks. Note that elements [a252] and [a261] are both present here and in the prompt in Fig. 9 as these are from the same observation.

# C    Task details

We show examples of goals given to the agent in the chat window under the "L2" difficulty and the task description seen by the agent under the "L3" difficulty for all the tasks within each skill category in WorkArena++. We also follow up (§ C.6) by showing all the 'protocols' the agent can refer to when solving an "L3" difficulty task mimicking a real-world scenario faced by knowledge workers.

## C.1    Planning and Problem Solving (66 × 2 tasks)

We devise tasks that require the agent to solve common enterprise problems that require decision making while respecting constraints. The tasks require the agent to follow a global plan that dictates the problems, constraints, and the desired outcome.

### C.1.1    Workload balancing

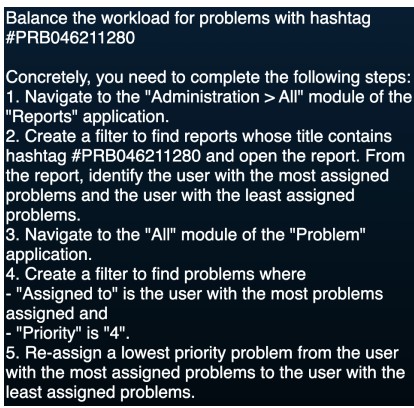
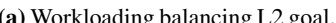
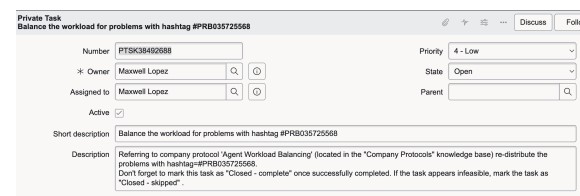

(a) Workloading balancing L2 goal.                (b) Workloading balancing L3 task description.

**Figure 11:** Workload balancing task: a) The goal given to the agent in chat for the L2 difficulty level. b) The description of the task assigned to the agent for the L3 difficulty level.

Redistribute non-uniformly assigned work across agents to make the workload balanced among them. We introduce more complexities based on the number of problems that need to be reassigned. In total, we have 3 workload balancing tasks across each L2 and L3 difficulty levels. We show the L2 goal and L3 task description in Fig. 11. The protocol for the task is in Fig. 23.

### C.1.2    Work assignment

Assign work (problems) across different categories such as software and hardware to relevant expert agents. We increase complexity by requiring the agent to assign critical problems based on their priority to more experienced agents under the different categories. In total, we have 6 work assignment tasks across each L2 and L3 difficulty levels. We show the L2 goal and L3 task description in Fig. 12 for a work assignment task requiring the agent to assign problems based on the expertise of the category experts. The protocol for the task is in Fig. 24.

### C.1.3    Scheduling requests

Schedule multiple problem requests based on constraints such as allowed time frame, overlap with other requests, impact and criticality of the problem, and allocated time to each request based on their risk. In total, we have 48 request scheduling tasks across each L2 and L3 difficulty levels. We show the L2 goal and L3 task description in Fig. 13 for a scenario where the requests are to be scheduled in an order based on their impact. The protocol for the task is in Fig. 25.

### C.1.4    Deduplicate problems

Mark problems that share the same problem statement as duplicates of each other. We introduce further complexity here for cases where the agent has to take into account the 'priority' of the problem while

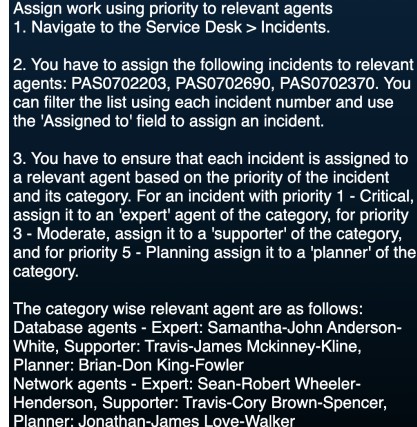

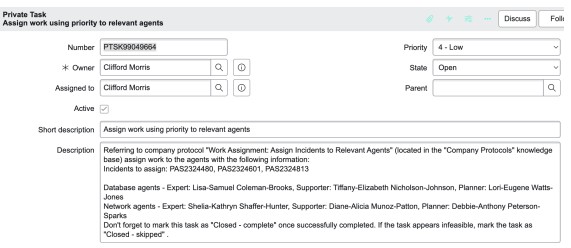

**(a)** Work assignment L2 goal.

**(b)** Work assignment L3 task description.

**Figure 12:** Work assignment task: a) The goal given to the agent in chat for the L2 difficulty level. b) The description of the task assigned to the agent for the L3 difficulty level.

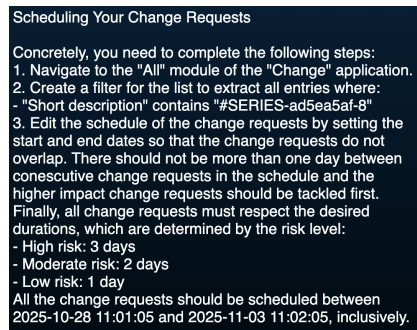

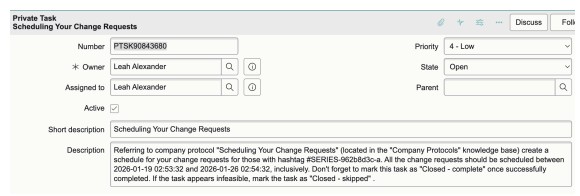

**(a)** Scheduling requests L2 goal.

**(b)** Scheduling requests L3 task description.

**Figure 13:** Scheduling requests task: a) The goal given to the agent in chat for the L2 difficulty level. b) The description of the task assigned to the agent for the L3 difficulty level.

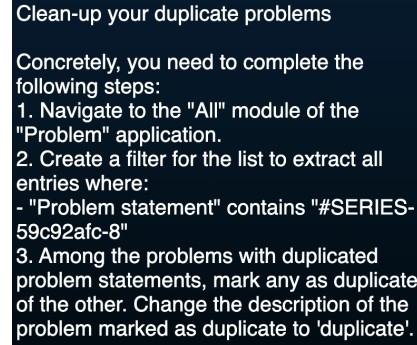

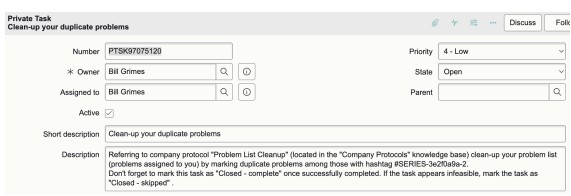

**(a)** Deduplicate problems L2 goal.

**(b)** Deduplicate problems L3 task description.

**Figure 14:** Deduplicate problems task: a) The goal given to the agent in chat for the L2 difficulty level. b) The description of the task assigned to the agent for the L3 difficulty level.

marking duplicates. In total, we have 9 problem deduplication tasks across each L2 and L3 difficulty levels. We show the L2 goal and L3 task description in Fig. 14. The protocol for the task is in Fig. 26.

## C.2 Information Retrieval (39 × 2 tasks)

To evaluate the retrieval ability of web agents, we formulate tasks requiring information extraction from visual elements such as plots and text elements such as forms. The agent is then requested to perform a diverse set of follow-up tasks using the retrieved information.

### C.2.1 Dashboard retrieval

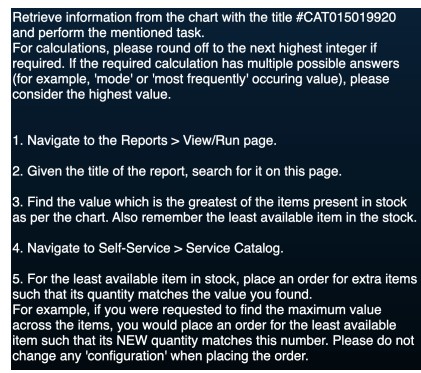

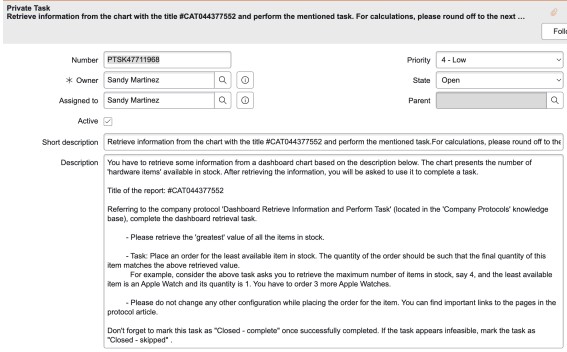

(a) Dashboard retrieval L2 goal.        (b) Dashboard retrieval L3 task description.

**Figure 15:** Dashboard retrieval task: a) The goal given to the agent in chat for the L2 difficulty level. b) The description of the task assigned to the agent for the L3 difficulty level.

Retrieve values or labels from bar plots and pie charts and use the information to perform subsequent tasks like filtering lists based on the values, creating new incidents using labels, order items low in stock, etc. In total, we have 29 dashboard retrieval tasks across each L2 and L3 difficulty levels. We show the L2 goal and L3 task description in Fig. 15 for a task requiring an agent to place order for items low in stock after looking at a catalog chart. The protocol for the task is in Fig. 27.

### C.2.2 List and form retrieval

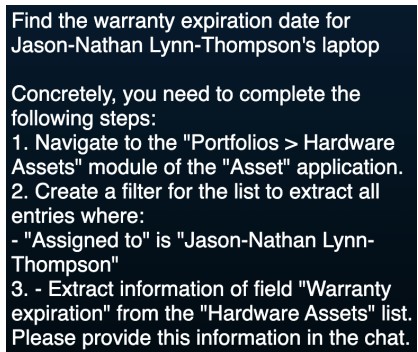

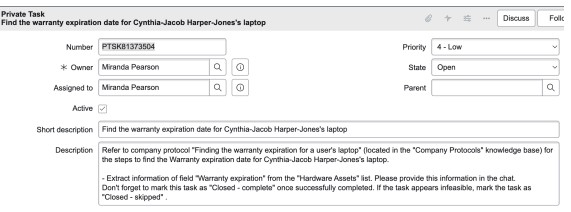

(a) List/ form retrieval L2 goal.        (b) List/ form retrieval L3 task description.

**Figure 16:** List/ form retrieval task: a) The goal given to the agent in chat for the L2 difficulty level. b) The description of the task assigned to the agent for the L3 difficulty level.

Retrieve information using filters and forms on the platform and perform subsequent tasks such as ordering items or providing an answer in the chat. In total, we have 10 dashboard retrieval tasks across each L2 and L3 difficulty levels. We show the L2 goal and L3 task description in Fig. 16 for a task requiring extracting information from a form. The protocol for the task is in Fig. 28.

## C.3 Data-driven decision making and reasoning (156 × 2 tasks)

We evaluate the ability of web agents to interpret data, perform logical and mathematical reasoning, and take task-related decisions.

### C.3.1 Expense management

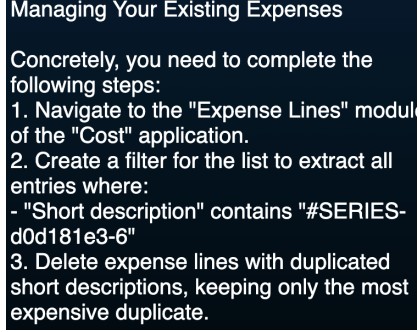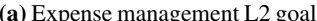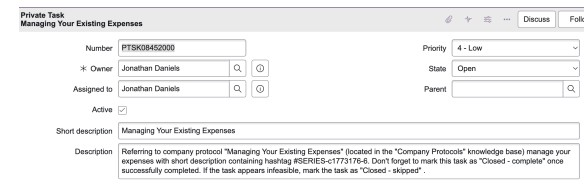

**(a)** Expense management L2 goal.

**(b)** Expense management L3 task description.

**Figure 17:** Expense management task: a) The goal given to the agent in chat for the L2 difficulty level. b) The description of the task assigned to the agent for the L3 difficulty level.

Review expenses to identify those with duplicated descriptions and delete the duplicated expenses according to a hierarchical set of rules. The agent may have to identify the expense to delete based on a date, an amount, or other attributes. In total, we have 12 expense management tasks across each L2 and L3 difficulty levels. We show the L2 goal and L3 task description in Fig. 17 for deleting expenses with duplicate descriptions while keeping the most expensive entry. The protocol for the task is in Fig. 29.

### C.3.2 Maximize investment

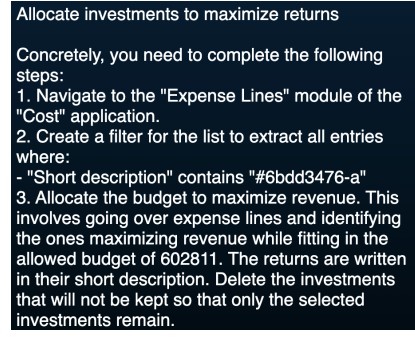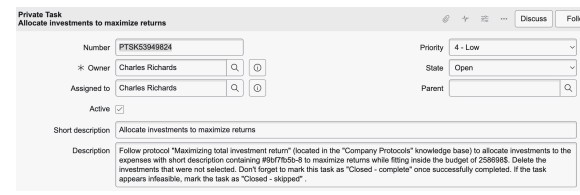

**(a)** Maximize investment L2 goal.

**(b)** Maximize investment L3 task description.

**Figure 18:** Maximize investment task: a) The goal given to the agent in chat for the L2 difficulty level. b) The description of the task assigned to the agent for the L3 difficulty level.

Select a subset of investments that maximize returns while staying within an allowed budget. We introduce complexities by requiring the agent to use different ways to solve the task such as deleting the expenses that will not be selected, providing the IDs of the selected investments, or answering with the total value of the selected investments in the chat. In total, we have 57 investment management tasks across each L2 and L3 difficulty levels. We show the L2 goal and L3 task description in Fig. 18 for an investment maximization task requiring the agent to delete expenses not being used. The protocol for the task is in Fig. 30.

### C.3.3 Mathematical computations based on charts

Perform global calculations such as mean, median, and mode over bar plots and pie charts and use them to perform subsequent tasks such as assigning problems to underperforming agents or ordering gifts for overperforming agents. In total, we have 87 tasks across each L2 and L3 difficulty levels requiring doing computations over plots. We show the L2 goal and L3 task description in Fig. 19 for a task requiring the agent to create 'probation' calls for all workers performing lower than the median performance. The protocol for the task is in Fig. 27.

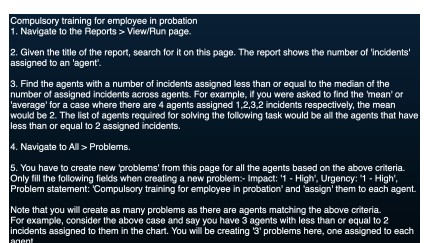

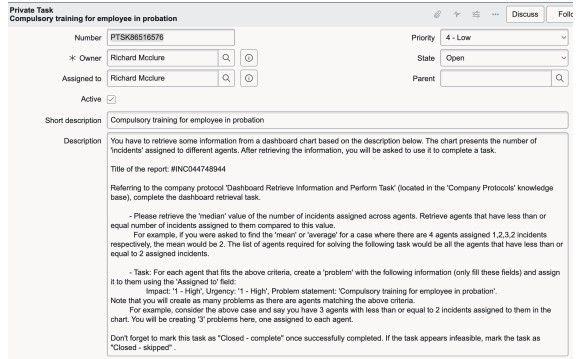

**(a)** Mathematical computations based on charts L2 goal.

**(b)** Mathematical computations based on charts L3 task description.

**Figure 19:** Mathematical computations based on charts task: a) The goal given to the agent in chat for the L2 difficulty level. b) The description of the task assigned to the agent for the L3 difficulty level.

## C.4 Sophisticated Memorization (29 × 2 tasks)

We explicitly test the ability of LLMs to memorize information necessary to behave as effective web-agents by devising tasks requiring following very clear yet information-intensive steps.

### C.4.1 Navigation + do

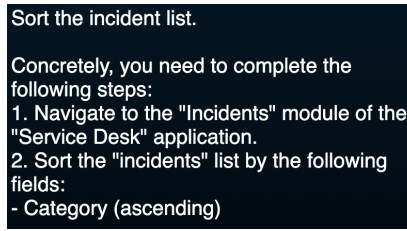

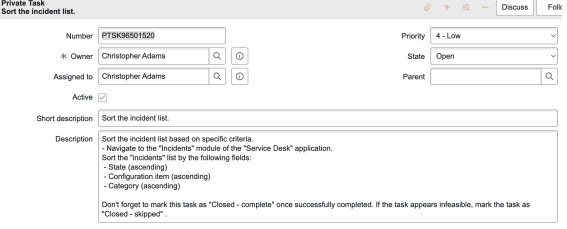

**(a)** Navigation + do L2 goal.

**(b)** Navigation + do L3 task description.

**Figure 20:** Navigation + do task: a) The goal given to the agent in chat for the L2 difficulty level. b) The description of the task assigned to the agent for the L3 difficulty level.

Navigate to a particular page on the platform and perform an aforementioned task. The agent has to remember details to navigate to the required page and details required to perform tasks like filling forms, sorting and filtering lists, and ordering items. In total, we have 26 navigation-and-do tasks across each L2 and L3 difficulty levels. We show the L2 goal and L3 task description in Fig. 20 for a task requiring an agent to navigate to the relevant page and perform a sort operation using the given details.

### C.4.2 User management

The agent has to onboard and offboard users by memorizing information. Onboarding users involves creating a user profile, ordering devices, and assigning them to the user created. Offboarding user involves deleting assigned hardware and marking user inactive. In total, we have 2 user management tasks across each L2 and L3 difficulty levels. We show the L2 goal and L3 task description for offboarding a user in Fig. 21. The protocol for the onboard user task is in Fig. 31 and the offboard user task is in Fig. 32.

## C.5 Contextual understanding through infeasible tasks (52 × 2 tasks)

The agent must understand that some instructions are impossible to perform (e.g., filling a form field which does not exist, ordering an invalid item configuration), and notify the user using the chat. We show the L2 goal and L3 task description in Fig. 22.

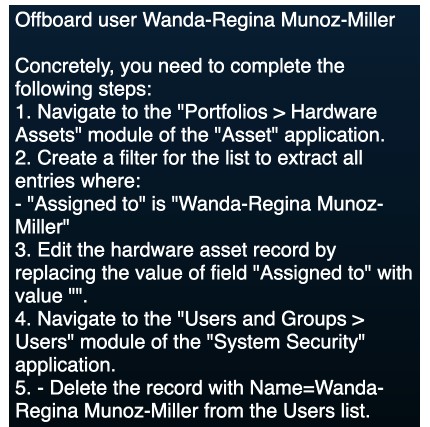

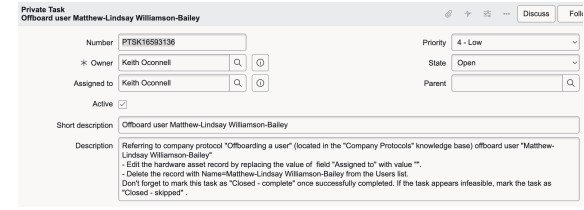

(a) Offboard user L2 goal.

(b) Offboard user L3 task description.

**Figure 21:** Offboard user (User management) task: a) The goal given to the agent in chat for the L2 difficulty level. b) The description of the task assigned to the agent for the L3 difficulty level.

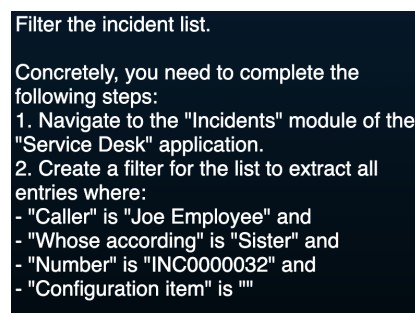

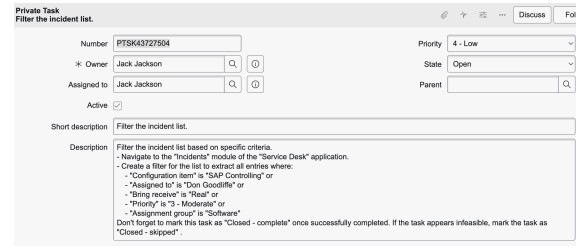

(a) Infeasible L2 goal.

(b) Infeasible L3 task description.

**Figure 22:** Infeasible task: a) The goal given to the agent in chat for the L2 difficulty level. b) The description of the task assigned to the agent for the L3 difficulty level.

## C.6   Task Protocols

We present all the protocols in our knowledge base that the agent can use to solve tasks in the L3 difficulty level. This makes the task very similar to how knowledge workers would refer to company documents in real-world settings.

# Agent Workload Balancing

👤 Authored by System Administrator  •  🗓 9d ago  •  👁 7 Views

## Agent Workload Balancing

### Introduction

Agent Workload Balancing is a process designed to distribute work evenly among agents by re-assigning tasks. The problems to re-distribute all contain specific hashtags in their problem statements. By balancing the workload, we ensure that no single agent is overwhelmed while others have fewer tasks. All problems can be found in the problem list.

### Steps for Agent Workload Balancing

#### 1. Identify Busiest and Least Busy Users

1. Among the problems with descriptions containing the required hashtag, identify the user with the most assigned problems and the user with the least assigned problems.

2. This information can be found in the report named 'Problems with hashtag {hashtag_name}'.

You can access the list of reports here.

#### 2. Find a Low Priority Problem

1. Locate a problem with the lowest priority (priority=5) that includes the appropriate hashtag in the problem statement and is assigned to the busiest user.

You can filter the problem list to find such a problem.

#### 3. Re-assign the Problem

1. Re-assign the identified low priority problem to the least busy user.

### Conclusion

Following these steps will help balance the workload among agents by re-assigning low priority problems from the busiest to the least busy user. For any issues or additional assistance, please contact the IT department.

---

This document serves as a guide to ensure effective workload management among agents within the organization.

---

**Figure 23:** Workload balancing task protocol.

# Work Assignment: Assign Incidents to Relevant Agents

## Introduction

This document outlines the process for assigning incidents to relevant agents based on both the category and the priority of the incident. Proper assignment ensures that incidents are handled by the most appropriate agents, leading to quicker and more effective resolutions. Follow these steps to assign incidents correctly.

## Steps for Assigning Incidents

### 1. Locate the Incident

1. Go to the incidents page and search for the incident using the incident number that needs to be assigned.

You can access the incidents page here.

### 2. Identify the Incident Category and Priority

1. Look at the category and priority of the incident. The category will be one of the following: 'Hardware', 'Software', 'Network', or 'Database'.
2. Assign the incident to the category experts mentioned in the task description.

### 3. Assign the Incident

1. If applicable, assign the incident to the relevant agent based on the priority of the incident. The priority levels are as follows:
   - **Priority 1 - Critical:** Assign to the 'expert' of the category.
   - **Priority 3 - Moderate:** Assign to the 'supporter' of the category.
   - **Priority 5 - Planning:** Assign to the 'planner' of the category.

## Conclusion

By following these steps, you can ensure that incidents are assigned to the most qualified agents based on their category and priority. This approach leads to faster resolution times and improved efficiency in incident management. For any issues or additional assistance, please contact the IT department.

This document serves as a guide to ensure proper assignment of incidents to relevant agents within the organization.

**Figure 24:** Work assignment task protocol.

# Scheduling Your Change Requests

## Scheduling Your Change Requests

### Introduction

Scheduling change requests is a critical process in managing IT infrastructure and operations. This process involves organizing and prioritizing change requests to ensure they are implemented efficiently and with minimal disruption to ongoing operations. Change requests often include updates, fixes, or enhancements that need to be scheduled within specific time blocks. The following guidelines outline how to create a valid schedule for change requests. As a first step, we note that change requests are grouped by hashtags placed in their short description. In general, creating a schedule is done for change requests that share a given hashtag and by setting the "Planned start date" and "Planned end date" for these change requests. Make sure to leave empty the "Actual start date" and "Actual end date" as they should only be filled when the work has started. All change requests can be found here.

### Scheduling Rules

#### 1. Time Constraints

1. All change requests must be scheduled within the allowed time frame. They should not start before the start of the schedule block and must end before it closes. Moreover, they should not overlap.

#### 2. Impact Order

1. Higher impact change requests must be scheduled before lower impact ones, ensuring the schedule prioritizes more significant changes.

#### 3. Consecutive Requests

1. Generally, there should be no more than one day between consecutive change requests.

2. If the change requests are required to follow a "tight" schedule, there should be no more than one hour between consecutive requests.

#### 4. Duration Compliance

1. All change requests must respect the desired durations, which are determined by the risk level:
   - High risk: 3 days
   - Moderate risk: 2 days
   - Low risk: 1 day

### Conclusion

Following these guidelines will help you create an efficient and effective schedule for your change requests, minimizing disruptions and prioritizing higher impact changes. For any issues or additional assistance, please contact the IT department.

This document serves as a guide to ensure proper scheduling and management of change requests within the organization.

**Figure 25:** Scheduling requests task protocol.

# Problem List Cleanup

## Problem List Cleanup

### Introduction

Problem List Cleanup ensures that the problem list remains organized and free from redundancy. In the problem list, some problem's "Problem statement" field contain a hashtag so we know which problem group they belong to. The problem list cleanup involves identifying problems with duplicated problem statements and marking them as duplicates.

### Steps for Problem List Cleanup

#### 1. Identify Duplicated Problems

1. Among the problems with problem statements containing the required hashtag, identify those with duplicated problem statements. This can be done by filtering the problem list .

#### 2. Mark Duplicated Problems

1. If two problems have the same priority, you can mark either one as a duplicate of the other. Otherwise, mark the lowest priority problem as a duplicate of the highest priority one to keep the highest priority problems open. A problem's priority can range from 1 (Critical) to 5 (Planning), with 1 being the highest priority.

2. Marking a problem as duplicate is done by opening the problem record and Clicking "Mark Duplicate".

#### 3. Additional Handling for Critical Priority Problems

1. If you need to mark a priority 1 (Critical priority) problem as a duplicate of another, change the description of this problem to "Duplicate" to avoid further confusion.

### Conclusion

Following these steps will help maintain a clean and efficient problem list by properly marking and managing duplicated problems. For any issues or additional assistance, please contact the IT department.

---

This document serves as a guide to ensure effective problem management within the organization.

**Figure 26:** Deduplicate problems task protocol.

# Dashboard Retrieve Information and Perform Task

## Dashboard Retrieve Information and Perform Task

### Introduction

Dashboards consists of either bar plots or pie charts containing important information, generally as a pair of some text and numbers. You would be asked to retrieve some information from the chart, which might be retrieving values such as the maximum and minimum value or requiring standard mathematical calculations such as mean, median, and mode. You may also require to extract the text associated with these values, such as the name of an agent. Once they are retrieved, you would be requested to perform a task using this information. Please note that we round off everything to the next *highest* integer and treat it as the 'value' for the task. For calculations having more than one possible answer (such as 'mode' or 'most frequent' values), we consider the greatest number as the value or information.

### Steps for performing dashboard retrieval tasks

#### 1. Retrieve information from the dashboard

1. Go to Reports > View/Run in the all menu. You can go to the page here as well.

2. Use the hashtag given in the task description to filter out the required dashboard chart and navigate to it.

3. Based on the task description, you will be given a 'value' field. This would mention if you need to fetch the minimum or maximum valued information or the mean, median, or the mode.

4. In certain cases, the description will also mention if you need to perform the following task by considering 'greater than or equal to' or 'lesser than or equal to' values for the retrieved information.

This is the retrieved information from the dashboard.

#### 2. Perform the task

After retrieving the information, you would be given a task to solve. Depending on the task requirements, you may need to perform one or more of the following:

- **Create form entries:** Fill forms with the given information. Only fill the fields mentioned in the task description.

- **Filter lists:** Filter the lists using the attribute mentioned in the task description and following the retrieved information.

- **Order items:** You would need to go to the service catalog and order an item after making some calculations of the quantity as per the task description.

The following links may be helpful for achieving the above tasks:

- Create incidents, problems or item requests

- Lists: asset list, hardware list, incident list, user list

- Service catalog for ordering items

### Conclusion

Following these steps will help you achieve the task of retrieving information from a dashboard and using it to solve another task.

This document serves as a guide to ensure proper management and allocation of investments within the organization.

**Figure 27:** Dashboard retrieval task protocol.

# Finding the warranty expiration for a user's laptop

● Authored by System Administrator   •   📅 9d ago   •   👁 10 Views

# Finding the Warranty Expiration for a User's Laptop

## Introduction

This document outlines the procedure for finding the warranty expiration date of a user's laptop. Ensuring that the warranty information is up-to-date is crucial for managing hardware assets and planning for replacements or repairs.

## Steps to Find the Warranty Expiration Date

### 1. Locate the Laptop Assigned to the User

1. Find the laptop assigned to the user by looking at the "Assigned to" field in the hardware asset list.

### 2. Find the Laptop's Warranty Expiration Date

1. Once the laptop is located, check the warranty expiration date on the asset details page.

## Conclusion

Following these steps will help you accurately find the warranty expiration date of a user's laptop. For any issues or additional assistance, please contact the IT department.

---

This document serves as a guide to ensure proper management of hardware warranties within the organization.

**Figure 28:** List/ form retrieval task protocol.

# Managing Your Existing Expenses

👤 Authored by System Administrator  •  📅 9d ago  •  👁 14 Views

## Managing Your Existing Expenses

### Introduction

Managing your existing expenses is a crucial aspect of maintaining financial accuracy and compliance within the company. This process involves reviewing expense lines to ensure no duplicates remain. Expense lines can be found here. Duplicate expenses can lead to inaccuracies in financial reporting and budget management, making it essential to handle them according to established guidelines. Expense lines normally contain specific hashtags so they can be grouped. The process of filtering the duplicates is done for specific hashtags.

### Steps for Managing Expense Duplicates

#### 1. Identify Duplicate Expenses

1. Review expense lines with short descriptions containing the given hashtag.

2. Identify expense lines that have the same short description, which are considered duplicates.

#### 2. Apply Rules to Handle Duplicates

1. Among duplicates, keep only those linked to tasks (e.g., Change Request, Incident) and delete the others.

2. If all duplicates are linked to tasks, or none are linked to tasks, keep only the oldest one.

3. If all duplicates have the same date, keep only the most expensive one.

4. Finally, if all duplicates have the same cost, you can keep any of the expenses.

### Conclusion

Following these guidelines will help you manage your existing expenses effectively, ensuring compliance with company policies and maintaining accurate financial records. For any issues or additional assistance, please contact the finance department.

This document serves as a guide to ensure proper management of expenses within the organization.

**Figure 29:** Expense management task protocol.

# Maximizing total investment return

👤 Authored by System Administrator  •  📅 9d ago  •  👁 30 Views

## Maximizing Total Investment Return

### Introduction

Investment projects are filed as expense lines and are tagged with project IDs (hashtags) so they can be grouped. Expense lines can be found here. Maximizing total investment return involves reviewing these expense lines and selecting the ones that maximize revenue while fitting within the allowed budget. The investment's returns are specified in their short descriptions and their costs appear in their "Amount" field. Depending on the specifics of the task, you may be required to provide different types of information.

### Steps for Maximizing Total Investment Return

#### 1. Review Expense Lines

1. Identify expense lines with short descriptions containing the given hashtag.
2. Evaluate the returns based on the short description and costs based on the "Amount" field to determine their suitability based on the allowed budget.

#### 2. Provide Required Information

Depending on the task requirements, you may need to provide one or more of the following:

- **Total Return of Selected Investments Only:** If requested, provide only the total return of the selected investments in the chat.
- **Selected Investments Only:** If requested, provide the values for the "Number" field of the selected investments in the chat.
- **Selected Investments and Total Return:** If requested, provide the values for the "Number" field of the selected investments as well as their total return in the chat.
- **Investment Lines Cleanup:** If requested, delete the investments that will not be kept so that only the selected investments remain.

### Conclusion

Following these steps will help you allocate investments effectively, ensuring that the selected projects maximize revenue while adhering to the budget constraints. For any issues or additional assistance, please contact the finance department.

---

This document serves as a guide to ensure proper management and allocation of investments within the organization.

**Figure 30:** Maximize investment task protocol.

# Onboarding a new user

👤 Authored by System Administrator  •  📅 9d ago  •  👁 9 Views

## Onboarding a new user

This document outlines the procedure for onboarding a new user within the company. Proper onboarding ensures that new users have the necessary tools and access to start their roles effectively and efficiently. Follow the steps below to complete the onboarding process.

## Steps for Onboarding a New User

### 1. Create a User Account

1. Access the user creation interface.
2. Enter the required information for the new user.
3. Save the new user account.

You can create a user account here.

### 2. Order a Laptop

1. Access the service catalog.
2. Order an Apple MacBook Pro 15" laptop for the new user.
3. Confirm the order details and submit the request.

You can order the laptop here.

### 3. Create a Hardware Asset

1. Access the hardware asset management interface.
2. Create a new hardware asset with the required configuration.
3. Assign the newly created hardware asset to the user.

You can create and assign a hardware asset here.

## Conclusion

Following these steps will ensure that new users are properly onboarded with the necessary equipment and access rights. For any issues or additional assistance, please contact the IT department.

---

This document serves as a guide to streamline the user onboarding process, ensuring consistency and efficiency across the organization.

**Figure 31:** Onboard user task protocol.

# Offboarding a user

👤 Authored by System Administrator • 📅 9d ago • 👁 12 Views

## Offboarding a user

### Introduction

This document outlines the procedure for offboarding a user within the company. Proper offboarding ensures that company assets are secured and that the departing user's access is properly removed. Follow the steps below to complete the offboarding process.

### Steps for Offboarding a User

#### 1. Un-assign Hardware Assets

1. Find the user's laptop by looking for the hardware asset assigned to them.
2. Un-assign the laptop by erasing the "Assigned to" field.

You can manage hardware assets in the hardware asset list.

#### 2. Delete the User Account

1. Access the user management interface.
2. Locate and delete the user account.

You can delete the user account here.

### Conclusion

Following these steps will ensure that departing users are properly offboarded and that all company assets and access rights are appropriately managed. For any issues or additional assistance, please contact the IT department.

---

This document serves as a guide to streamline the user offboarding process, ensuring consistency and security across the organization.

**Figure 32:** Offboard user task protocol.

# D   Evaluation Curriculum Details

Though WorkArena++ consists of 682 tasks, each task can be instantiated with different configurations controlled by a task seed to get a task 'instance'. For example, the task requiring to maximize investment returns can have a random value for the maximum investment allowed. This value is generated using a random generator initialized using a task-level seed. Hence, to make evaluation across numerous models and agents feasible and also have reproducibility across evaluations, we propose a standardized way to sample task instances given tasks across skill categories. We aim to design a curriculum that ensures we cover all the skills in our categorization while having uniformity across these categories.

We start by creating buckets of tasks across each skill category based on the requirement of novel skills and abilities to solve the task. For example, ordering an iPad would essentially require the same abilities for an agent as ordering a Macbook, given the same interface and interactions necessary. Hence, these tasks would fall under the same bucket. We thus create a hierarchical categorization of all the tasks starting with the skill categories ( §3.2) and dividing the tasks within each skill category (§ C) into buckets. Within a bucket, we sample tasks with weights ensuring maximum skill coverage and minimum redundancy. Each skill is sampled with different number of seeds to ensure uniformity across each category.

Our curriculum only requires one meta-seed (m) to sample fixed configurations to create task instances across all the tasks in each category. Consider each skill category has task "buckets" and a corresponding "weight" for each bucket. Then for each category to be sampled "num_seeds" number of times, our curriculum follows the algorithm to get the WorkArena++ "Benchmark" $\mathcal{T}$ for reproducible and uniform evaluation,

---

**Algorithm 1** Agent Curriculum for WorkArena++ Benchmark

---

CATEGORIES ← Skill categories with bucket, weight, and num_seeds information
m ← Meta seed for reproducibility
$\mathcal{T} = ()$ ← Tuple of (task, seed) comprising the task and its task seed to instantiate the task with a configuration

random_generator = random_generator_init(m)
**for** category **in** CATEGORIES **do**
   seeds = random_generator(CATEGORIES[category]["num_seeds"])
   task_buckets = CATEGORIES[category]["buckets"]
   task_bucket_weights = CATEGORIES[category]["weights"]
   **for** seed **in** seeds **do**
     random_bucket_generator = random_generator_init(seed)
     **for** task_bucket, task_bucket_weight **in** zip(task_buckets, task_bucket_weights) **do**
       tasks = random_bucket_generator.sample(task_bucket, task_bucket_weight, replace=False)
       **for** task **in** tasks **do**
         $\mathcal{T} \leftarrow \mathcal{T} + (\text{task}, \text{seed})$

return $\mathcal{T}$ ← WorkArena++ Benchmark

---

We reserve seeds (m) 0-9 for evaluation and the remaining seeds may be used for agent tuning. The "CATEGORIES" dictionary is shown in Fig. 33. Our curriculum yields 235 tasks for each L2 and L3 difficulties.

```python
AGENT_CURRICULUM = {
    "planning_and_problem_solving": {
        "buckets": [
            MARK_DUPLICATE_PROBLEMS_TASKS,
            WORKLOAD_BALANCING_TASKS,
            WORK_ASSIGNMENT_TASKS,
            SMALL_BASE_SCHEDULING_TASKS,
            LARGE_BASE_SCHEDULING_TASKS,
            SMALL_TIGHT_SCHEDULING_TASKS,
            LARGE_TIGHT_SCHEDULING_TASKS,
        ],
        "num_seeds": 2,
        "weights": [9, 3, 6, 1, 1, 1, 1],
    },
    "information_retrieval": {
        "buckets": [
            DASH_AND_ORDER,
            DASH_AND_CREATE_INCIDENT,
            DASH_AND_CREATE_PROBLEM,
            DASH_COMPUTE_MIN_FILTER_LIST,
            DASH_COMPUTE_MAX_FILTER_LIST,
            DASH_AND_REQUEST,
            WARRANTY_CHECK_TASKS,
            FIND_AND_ORDER_ITEM_TASKS,
        ],
        "num_seeds": 7,
        "weights": [1, 1, 1, 1, 1, 1, 1, 1],
    },
    "data_driven_decision_making_and_reasoning": {
        "buckets": [
            EXPENSE_MANAGEMENT_TASKS,
            MAXIMIZE_INVESTMENT_RETURN_TASKS,
            DASH_COMPUTE_MEAN_AND_ORDER,
            DASH_COMPUTE_MEDIAN_AND_ORDER,
            DASH_COMPUTE_MODE_AND_ORDER,
            DASH_COMPUTE_AND_CREATE_INCIDENT,
            DASH_COMPUTE_AND_CREATE_PROBLEM,
            DASH_COMPUTE_MEAN_FILTER_LIST,
            DASH_COMPUTE_MEDIAN_FILTER_LIST,
            DASH_COMPUTE_MODE_FILTER_LIST,
            DASH_COMPUTE_MEAN_AND_REQUEST,
            DASH_COMPUTE_MEDIAN_AND_REQUEST,
            DASH_COMPUTE_MODE_AND_REQUEST,
        ],
        "num_seeds": 1,
        "weights": [12, 28, 1, 1, 1, 3, 3, 1, 1, 1, 1, 1, 1],
    },
    "sophisticated_memory": {
        "buckets": [
            NAVIGATE_AND_CREATE_TASKS,
            NAVIGATE_AND_ORDER_TASKS,
            NAVIGATE_AND_FILTER_TASKS,
            NAVIGATE_AND_SORT_TASKS,
            OFFBOARD_USER_TASKS,
            ONBOARD_USER_TASKS,
        ],
        "num_seeds": 8,
        "weights": [1, 1, 1, 1, 1, 1],
    },
    "contextual_understanding_infeasible_tasks": {
        "buckets": [
            INFEASIBLE_NAVIGATE_AND_CREATE_WITH_REASON,
            INFEASIBLE_NAVIGATE_AND_CREATE,
            INFEASIBLE_NAVIGATE_AND_ORDER_WITH_REASON,
            INFEASIBLE_NAVIGATE_AND_ORDER,
            INFEASIBLE_NAVIGATE_AND_FILTER_WITH_REASON,
            INFEASIBLE_NAVIGATE_AND_FILTER,
            INFEASIBLE_NAVIGATE_AND_SORT_WITH_REASON,
            INFEASIBLE_NAVIGATE_AND_SORT,
        ],
        "num_seeds": 4,
        "weights": [1, 1, 1, 1, 1, 1, 1, 1],
    },
}
```

**Figure 33:** Agent curriculum for the WorkArena++ Benchmark.

# E    Expanded Features

## E.1    Application Themes

To add visual diversity, we created 10 themes of fictitious companies. They are all listed below.

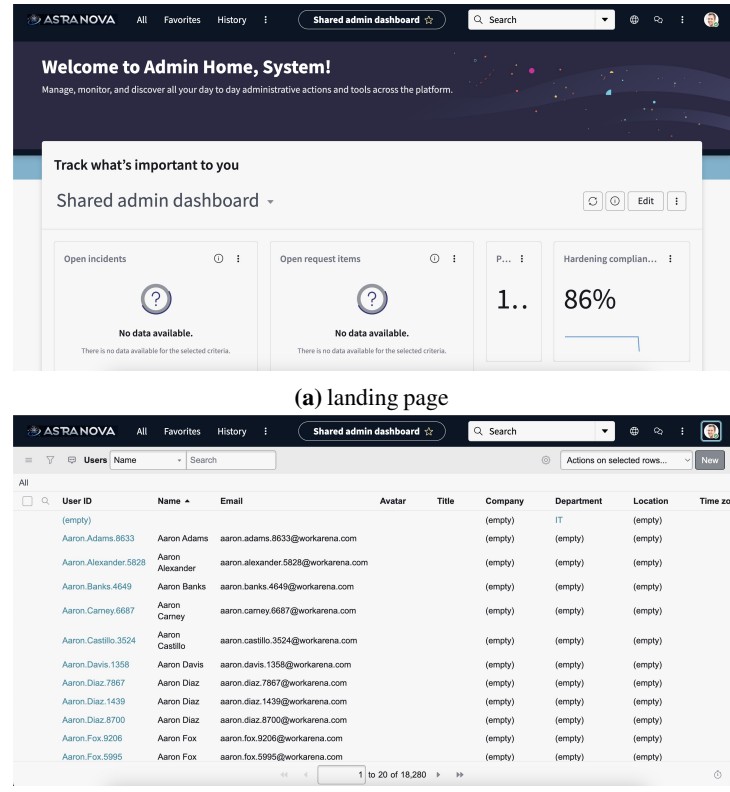

**(a)** landing page

**(b)** list view

**Figure 34:** Astranova theme

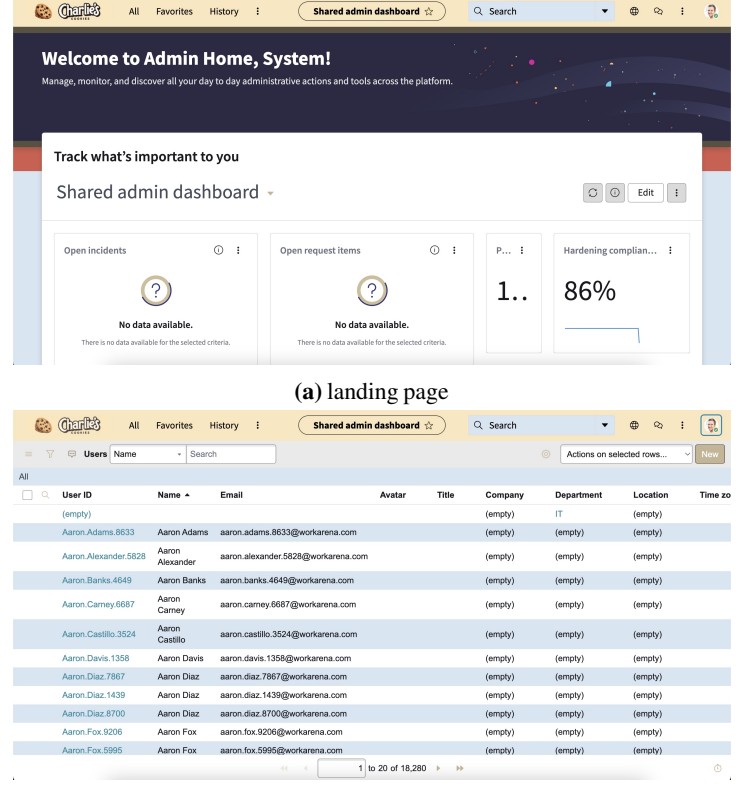

**(a)** landing page

**(b)** list view

**Figure 35:** Charlie's cookies theme

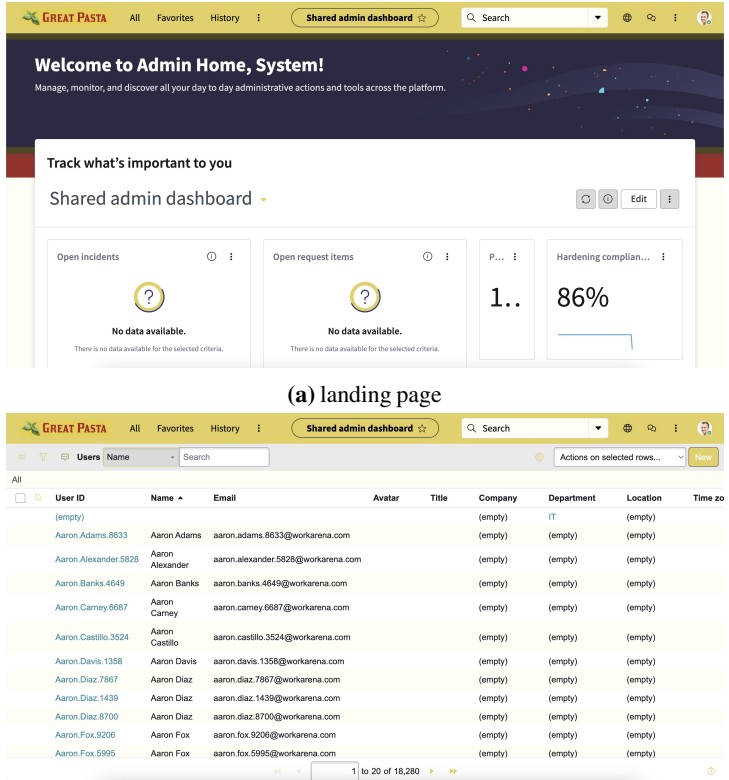

**(a)** landing page

**(b)** list view

**Figure 36:** Great Pasta theme

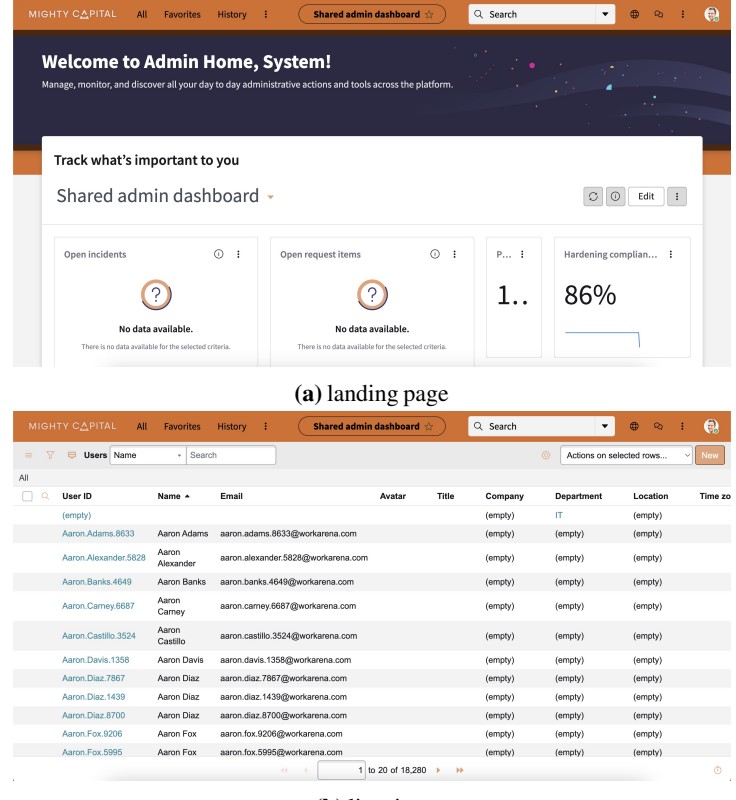

**(a)** landing page

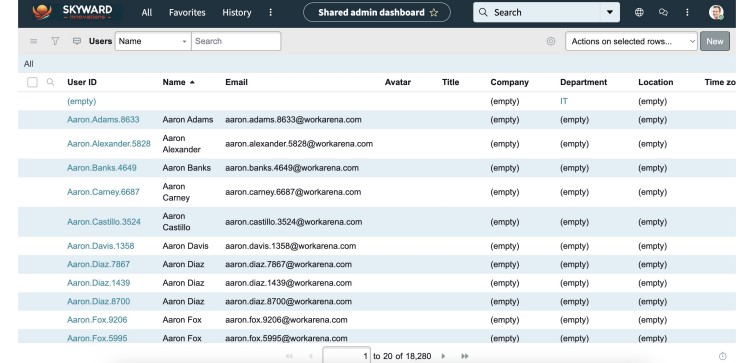

**(b)** list view

**Figure 37:** Mighty Capital theme

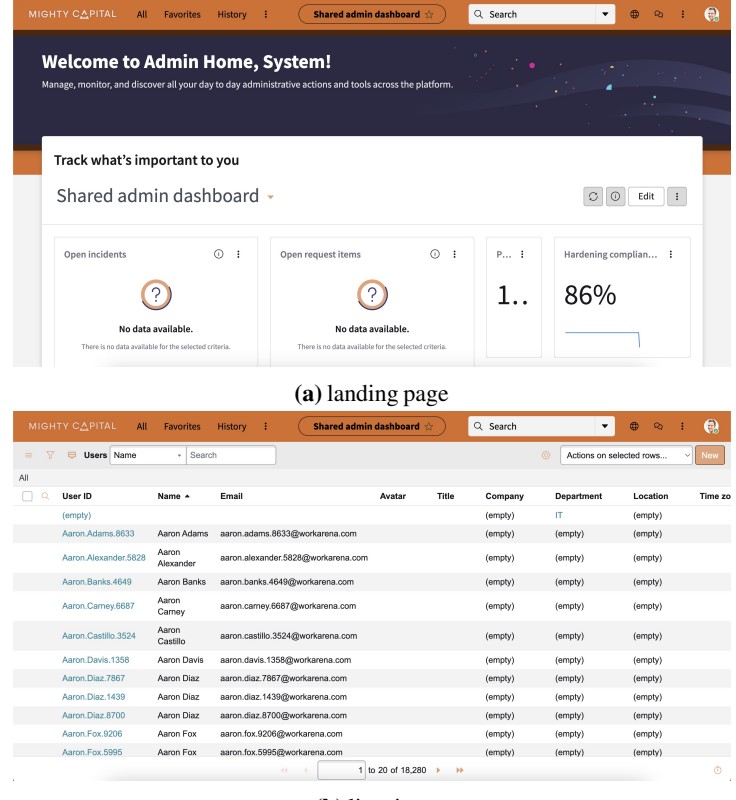

**(a)** landing page

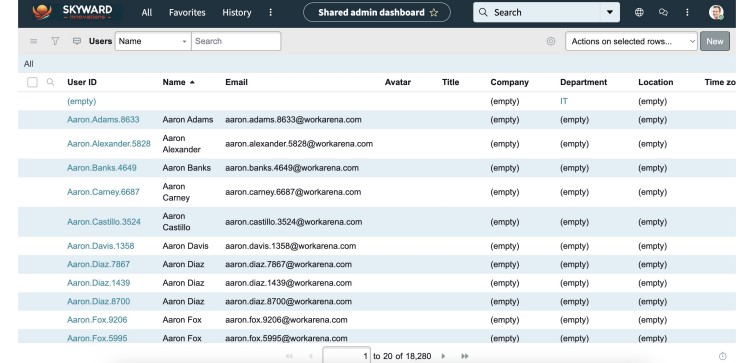

**(b)** list view

**Figure 38:** Skyward theme

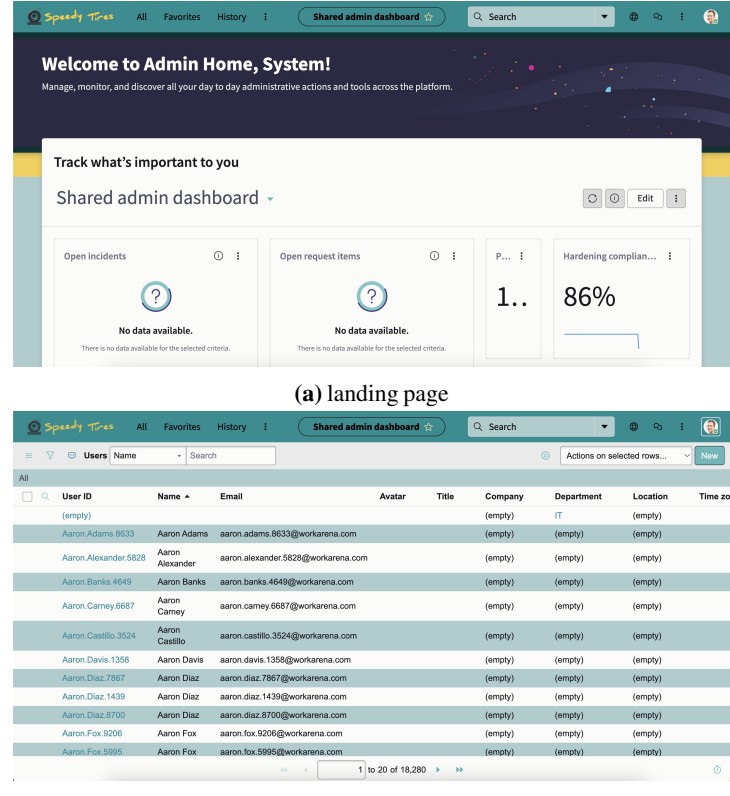

**(a)** landing page

**(b)** list view

**Figure 39:** SpeedyTires theme

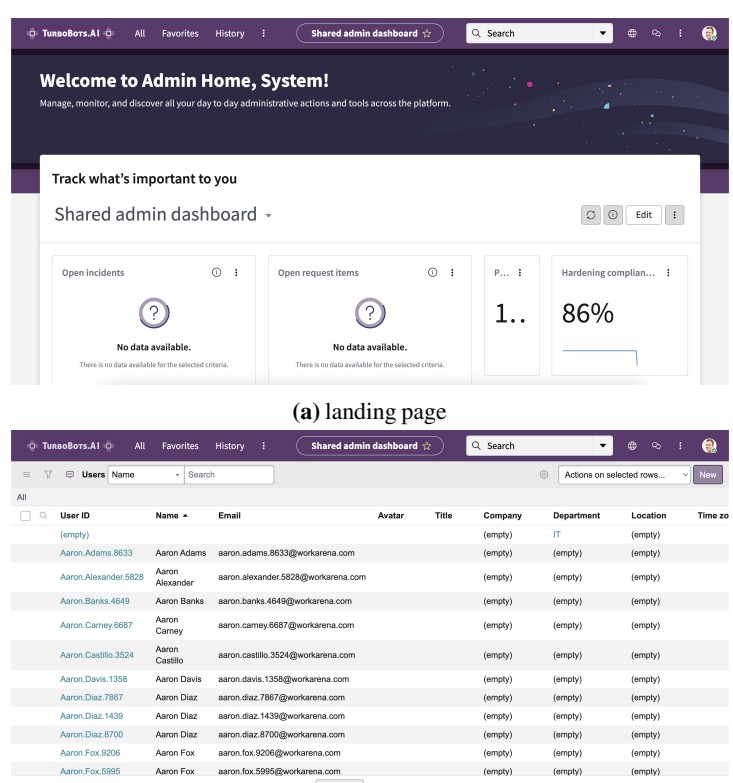

**(a)** landing page

**(b)** list view

**Figure 40:** TurboBots theme

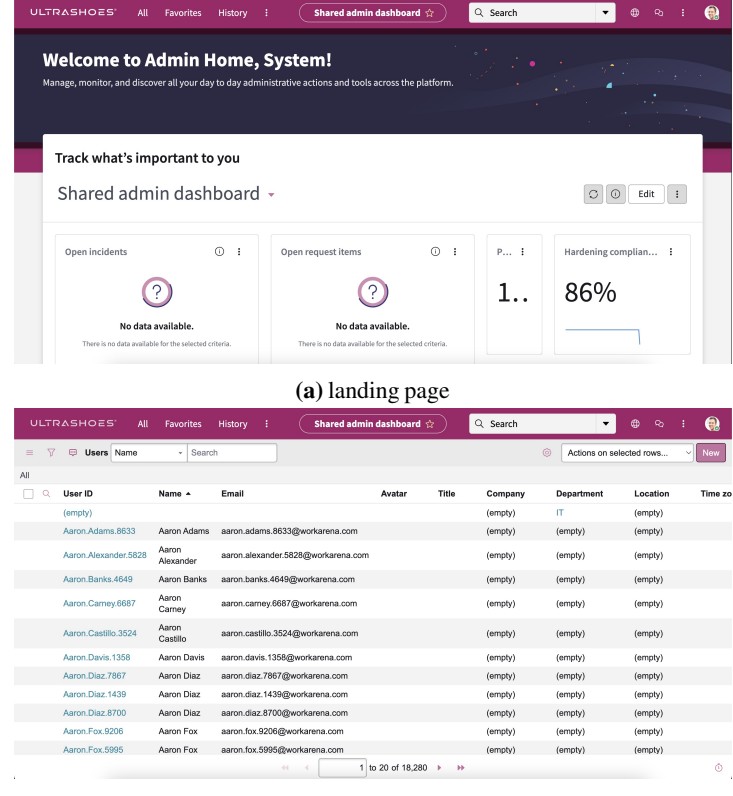

**(a)** landing page

**(b)** list view

**Figure 41:** UltraShoes theme

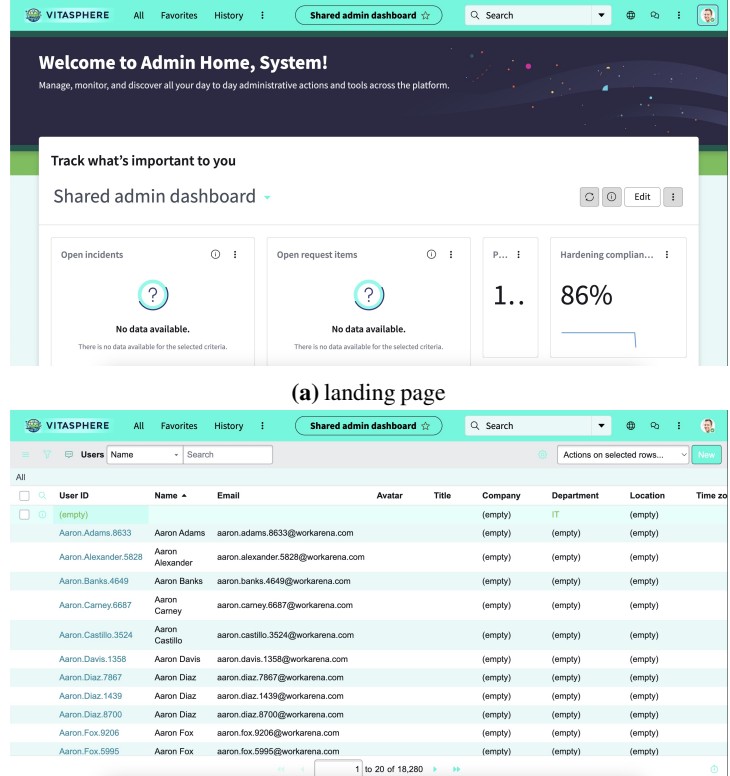

**(a)** landing page

**(b)** list view

**Figure 42:** VitaSphere theme

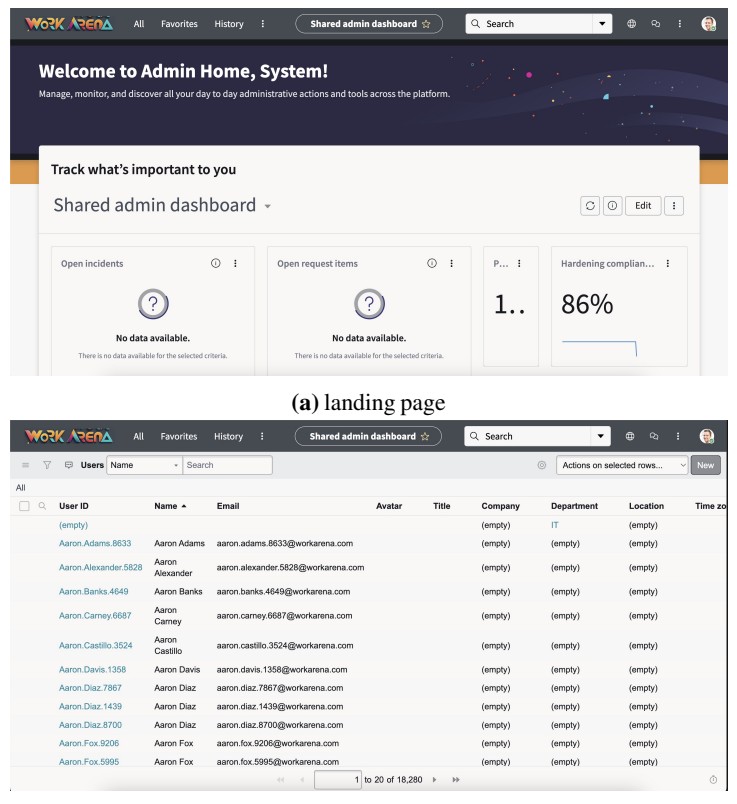

**(a)** landing page

**(b)** list view

**Figure 43:** WorkArena theme

## E.2   Extracting traces and designing more tasks

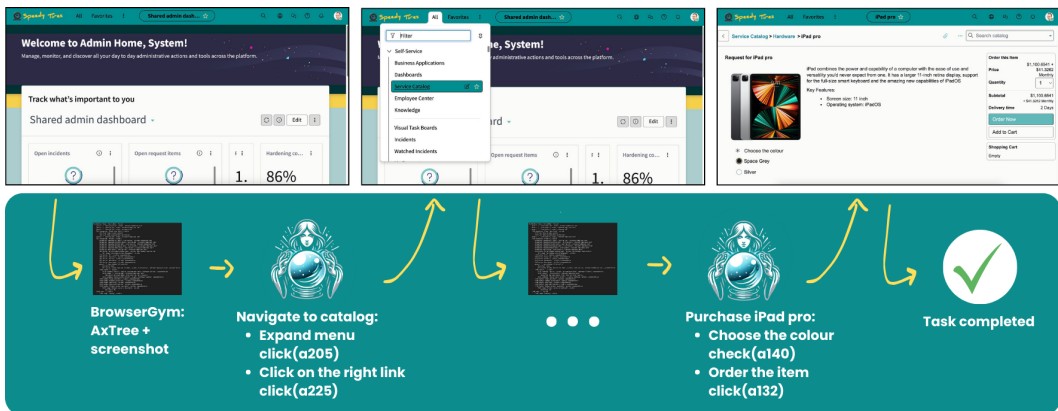

**Figure 44:** Extracting traces. In this example, the goal of the task is to order an iPad pro from the service catalog. In the first step, the task starts from the ServiceNow landing page. BrowserGym extracts a screenshot of this page + the page's accessibility tree. In step 2, the oracle function is used to navigate to the service catalog using Playwright code. This action is recorded and saved as a trace. Finally, in step 3 the oracle purchases an iPad pro by navigating to the item page and ordering it. All these actions are recorded and saved alongside the accessibility tree of the pages where the actions were performed.

**Extracting traces**   All tasks from WorkArena and WorkArena++ implement an "oracle" function that solves them step-by-step with Playwright code. To extract traces of solutions, we leverage BrowserGym, which extracts a screenshot of the page, the AxTree, the DOM as well as other optional features (e.g. set-of-marks) and we intercept the Playwright code outputted by our oracle function. Together, this allows us to generate traces for the resolution of tasks in WorkArena++. The process is illustrated in Fig. 44.

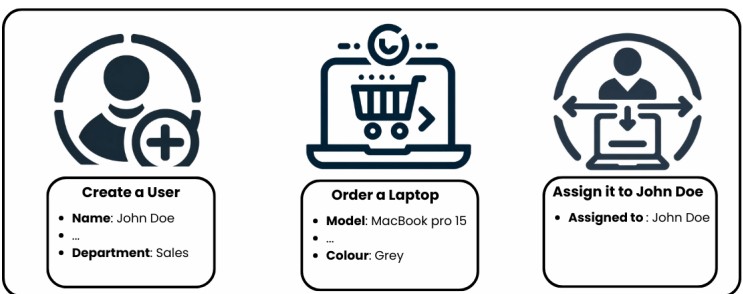

**Figure 45:** User OnBoarding task

**Designing new tasks**   All existing tasks in WorkArena can be combined to create new workflows. When chaining tasks to create compositional workflows, the individual tasks' oracle functions are called in succession to solve the task step-by-step. Moreover, the validation of these task can either be a sequential validation of the individual tasks or a global validation made by querying the database. This affords the possibility of combining low-level tasks to create realistic compositional tasks or modify existing ones to make them more complex. For example, the basic user onboarding task consists in creating a user, ordering a laptop for them and finally assigning the laptop to the user that was created. As this task is sequential, all of its individual steps are validated one by one. It is illustrated in Fig. 45.

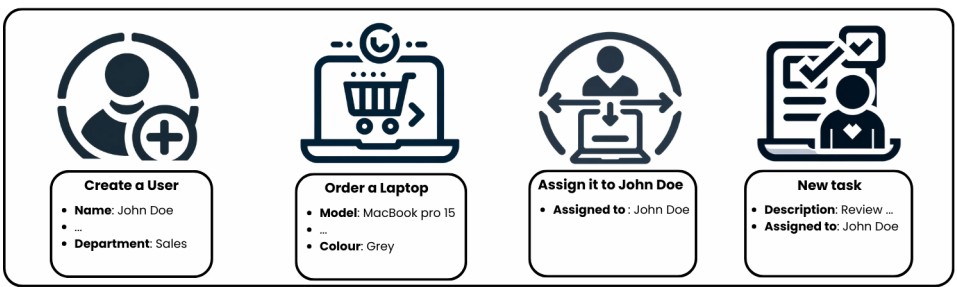

**Figure 46:** Extended User OnBoarding task

Suppose this workflow needs to be modified by adding a new step: Creating a first task assigned to the user. This can easily be done by adding a new task to the workflow and validating it. This is illustrated in Fig. 46.

# F  Error Analysis

In the following section, we will highlight some examples of errors made by our agents.

**Information Retrieval**

In BrowserGym, element in the page are assigned an identifier called bid, that can be used in the action functions. Normal element are assigned numbers, iframes are assigned letters, and elements within iframes are assigned the combination of the iframe bid and a number. For example, element '324' in iframe 'a' will have the bid 'a324'.

In the following example, the agent sometimes gets confused by the bid, and as in this example, forgets the iframe identifier. This error can have one of two consequences. The bid exists, and the model clicks on a random element of the web page. The bid does not exist, and the action causes an error, effectively not doing anything on the web page.

```
Axtree
...
[a252] link 'Hardware', clickable, visible
...
```

```
LLM Output
<think>
From the previous action, I clicked on the "Software" link.  Now, I need
to navigate to the "Hardware" section to find and order the iPad Pro.  I
can see the "Hardware" link with bid '252'.  I will click on this link
to proceed to the hardware store.
</think>
<action>
click('252')
</action>
```

This action causes an error that can be fed to the agent on the next step.

**Information Retrieval**

The model sometimes struggles with understanding the information that are available in the page. In the next example, it has to sort a table by two criterion, in a certain order. On this step, the table is already sorted by ascending 'Number'. The agent seems to see this information, as it is mentioned in the think section. However, the agent does not seem to understand it, as it keeps on clicking on the column header to sort it.

```
Goal
Goal: Sort the change request list.
Concretely, you need to complete the following steps:
```
1. Navigate to the "All" module of the "Change" application.
2. Sort the "change requests" list by the following fields:
   - **Number** (ascending)
   - **Close code** (ascending)

```
LLM Output
<think>
I need to sort the "change requests" list by the "Number" field in
ascending order.
I will click on the "Sorted in ascending order by Number" button.
```

```
</think>
<action>
click('1116')
</action>
```

The column header is indicated as 'Sorted in ascending order by Number', and yet the agent tries to sort it.

**Exploration**

The `all-menu` (Fig. 47) is a key component for navigating the WorkArena environment. It can sometimes be an obstacle for the agent, as it is composed of many foldable sub-menus.

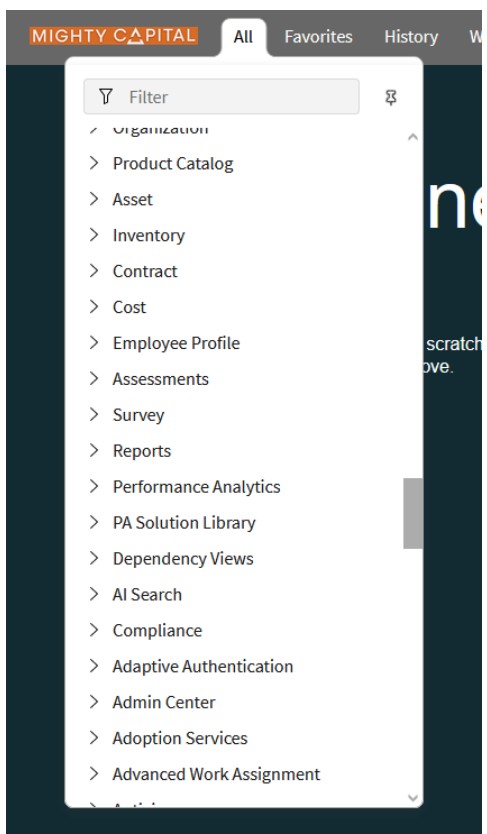

**(a)** Folded version of the `all-menu`

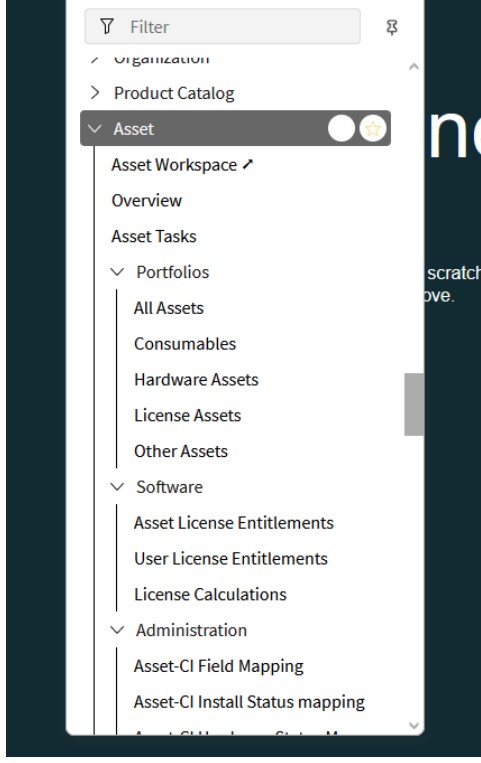

**(b)** Unfolded version of the `all-menu`

**Figure 47:** `all-menu`

**Goal**

Goal: Sort the hardware asset list.
Concretely, you need to complete the following steps:

1. Navigate to the *Portfolios > Hardware Assets* module of the *Asset* application.
2. Sort the "hardware" list by the following fields:
   - **Due in** (ascending)
   - **Model category** (ascending)
   - **Stockroom** (descending)

In the above example, as the *Asset* sub-menu is folded, its links are not available in the axtree. The agent needs to either lookup for *Hardware Assets* in the filtering search bar, of click on the *Asset* sub-menu, and then click on the *Hardware Assets* link that unfolds. The agent is sometimes not able to find those sub-menus and struggles to get to the right application.

**Exploration**

Another example of failed exploration occurs when filling forms. Forms have sections that require writing in information, which can be split into separate tabs in the form.

---

**Goal**

Goal: Create a new "Normal" change request
Concretely, you need to complete the following steps:

1. Navigate to the "All" module of the "Change" application.

2. Create a new change request with a value of "gigantean stageworthy plowable gagership schairerite" for field "Justification", a value of "training brutely brab galloptious acetanilide" for field "Risk and impact analysis", a value of "Paeonia taurocholic springmaking smattery Darren" for field "Test plan", a value of "midmonthly pimploe roosted uncowl nothingarian" for field "Implementation plan", a value of "Java Application Server change" for field "Short description", a value of "Java Application Server FLX" for field "Configuration item", a value of "3 - Low" for field "Impact", a value of "Other" for field "Category", a value of "CHG0000010" for field "Number", **a value of "cephalobranchiate resuscitant flashingly gauchely aclidian" for field "Close notes"**, and a value of "webfooter parapraxia Typhoean bockeret Labrador" for field "Backout plan".

---

In Fig. 48, the agent has filled all but one section of the form. The remaining section 'Close notes' is in the 'Closure Information' tab, and the agent needs to switch to the tab before being able to fill this last section. However, it decides to fill another section, overwriting old content (Fig. 49).

**Action Consequences Assessment - Hallucinated Consequences**

Interestingly, from Fig. 49, the error compounds as the model considers its job done, and proceeds to submit the form as is, giving the following thought/action:

---

**LLM Output**

```
<think>
I have already filled in most of the required fields for the new
change request.  The only remaining action is to submit the form.  The
"Submit" button is identified by bid 'a1229'.  I will click this button
to complete the creation of the new change request.
</think>
<action>
click('a1229')
</action>
```

---

This ends the run, causing the task to fail as the form is not filled the correct way.

**Hallucinations - Made Up Actions**

The model occasionally comes up with functions that would help it complete the task. Typically, the model has sometimes used functions like `check` and `uncheck`, that are not available in the highlevel action set provided to the model. Those could possibly be reminiscences of playwright data during the model pre-training that would interfere with our action space.

In the same way, the model sometimes tries to shortcut the function inputs, especially by replacing the bid by its target's name, causing an error:

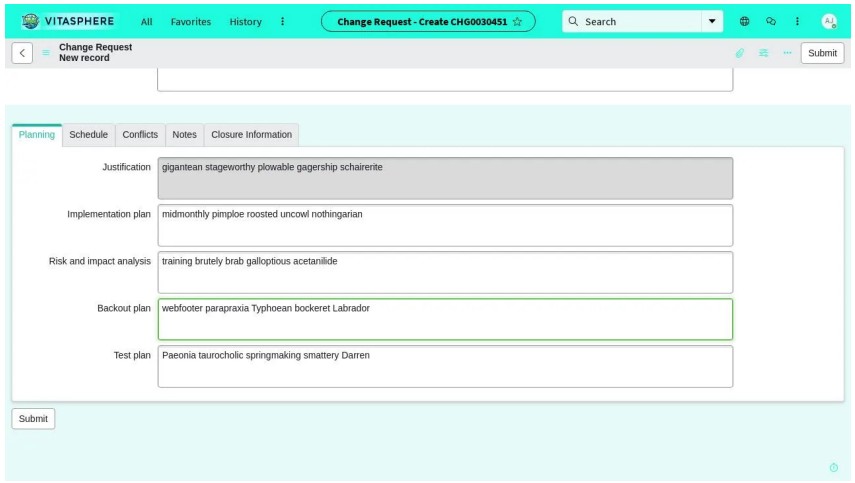

**Figure 48:** Request form on track to be fully filled

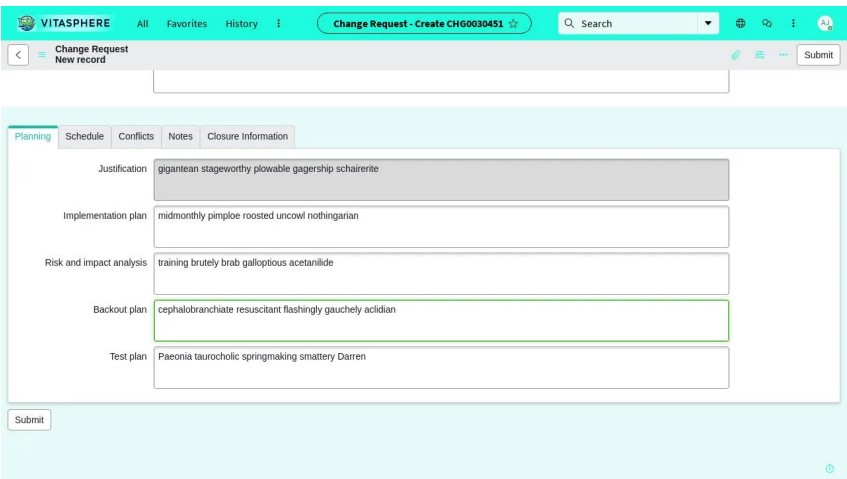

**Figure 49:** The agent fills the wrong section of the form with the last piece of information

**LLM Output**

```
<think>
I am currently on the "Service Catalog" page.  I need to navigate to
the hardware store and order 5 "iPad mini" with configuration 'Choose
the colour':  'Purple', 'Choose the storage':  '64'.  I have already
searched for "iPad mini" in the search catalog field and clicked on the
"iPad mini" link to access the hardware store.  Now, I will select the
"Purple" color option.
</think>
<action>
select_option('colour', 'Purple')
</action>
```

## Hallucinations - Imaginary Buttons

When stuck, the model sometimes tries to cheat using buttons that gives the solution to the task, or a part of it:

```
<think>
To navigate to the "Service Catalog" module of the "Self-Service"
application, I need to explore the available menus.  The "All" button
seems to be the main menu where I can find different modules.  I will
start by expanding the "All" menu to look for the "Service Catalog"
option.
</think>
<action>
click('button:has-text("All")')
</action>
```

Here the model makes up a button that would find the correct bid automatically, which causes an error.

**Hallucination - Asking for help**

The L3 added complexity can sometimes cause the agent to not understand why the task is not completed. In the following example, the model wrongly changed the form's 'State' field to 'Closed Complete'. It then hallucinates the task to be complete, and believes there should be done, and asks for confirmation from the user.

```
<think>
The task is to order a standard laptop from the service catalog and
mark the task as "Closed - complete" once successfully completed.  The
"State" field was set to "Closed Complete" and the "Update" button was
clicked to save the changes.
To confirm that the task has been successfully marked as completed,
I need to verify the current value of the "State" field.  Since the
"State" field was already set to "Closed Complete" before, I will
now check if there are any further actions required or if there is a
confirmation message indicating that the task has been successfully
completed.
I will inspect the "State" field to ensure it is set to "Closed
Complete" and look for any confirmation messages or changes in the form
that indicate the task has been successfully updated.
</think>
<action>
send_msg_to_user('The task has been marked as "Closed - complete".
Please verify if there are any further actions required or if there is
a confirmation message indicating that the task has been successfully
completed.')
</action>
```

**Goal Understanding - Thought/Action Consistency**

In the following task:

Goal: Order an iPad Mini from the service catalog
Concretely, you need to complete the following steps:

1. Navigate to the "Service Catalog" module of the "Self-Service" application.

2. Go to the hardware store and order 5 "iPad mini" with configuration 'Choose the colour': 'Purple', 'Choose the storage': '64'

The agent at some point performs the following action:

```
<think>
I am currently on the "Belkin iPad Mini Case" page.  I need to navigate
to the hardware store and order 5 "iPad mini" with configuration
'Choose the colour':  'Purple', 'Choose the storage':  '64'.  I will
click on the "Order this Item" button with bid 'a237' to proceed with
the order.
</think>
<action>
click('a237')
</action>
```

In this example the model orders an iPad case, instead of the requested iPad Mini. Ordering is considered the end of the task in WorkArena, causing this run to fail. The model seems inconsistent with himself, as it asserts it needs to "navigate to the hardware store", but proceeds to order the case anyways.

**Goal Understanding - Low-Level Understanding in L3**

L3 task come with the added complexity that the task is described in a ticket. Sometimes the agents do not understand that this is a their task, and see the information in the tickets as fulfilled tasks. For example, in Fig. 50, the agent fails to understand that the information it is reading is the task at hand.

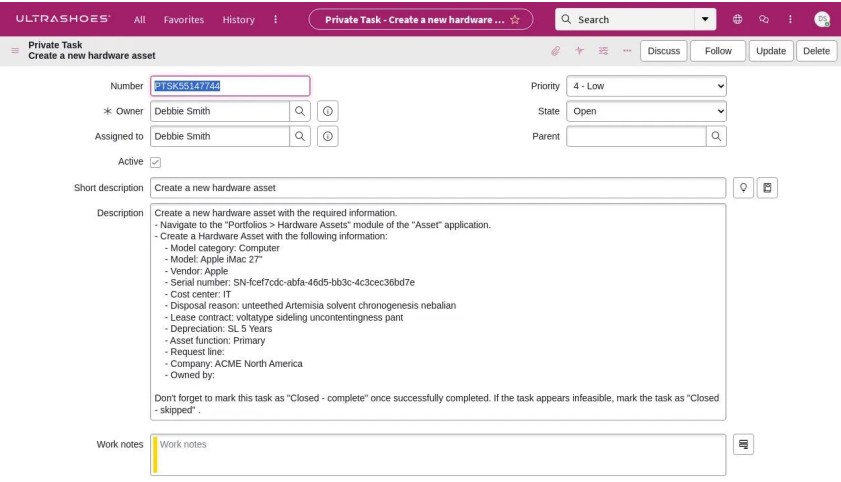

**Figure 50:** Task ticket from L3

This results in the action:

**LLM Output**

```
<think>
The current page is a ServiceNow interface with a task to create a new
hardware asset.  The task details are visible, and the form fields are
already filled with the required information.  The next logical step
is to mark the task as "Closed - complete" since the task appears to be
successfully completed.
To do this, I need to change the "State" field from "Open" to "Closed
Complete".  The "State" field is a combobox with bid 'a270'.  I will
select the option "Closed Complete" from the dropdown.
</think>
<action>
select_option('a270', 'Closed Complete')
</action>
```

The agent mistakes the information and considers the task done, closing the ticket.

**Action Consequences Assessment - Hallucinated Consequences**

Because of other errors, agents can be lead to think they have successfully completed parts of the tasks. Typically, when completing forms, lack of exploration can lead them to fill information in the wrong fields. However, instead of correcting those errors, they will go on and consider this part of the task done. The example below illustrates this error:

---

**Goal**

**Goal:** Create a new hardware asset
Concretely, you need to complete the following steps:

1. Navigate to the "Portfolios > Hardware Assets" module of the "Asset" application.

2. Create a new hardware asset with a value of "Computer" for field "Model category", a value of "Apple iMac 27"" for field "Model", a value of "Apple" for field "Vendor", a value of "SN-566be2cf-3f63-4e26-8aa5-54ad26c07875" for field "Serial number", a value of "IT" for field "Cost center", a value of "unteethed Artemisia solvent chronogenesis nebalian" for field "Disposal reason", a value of "voltatype sideling uncontentingness pant" for field "Lease contract", a value of "SL 5 Years" for field "Depreciation", a value of "Primary" for field "Asset function", a value of "" for field "Request line", a value of "ACME North America" for field "Company", and a value of "" for field "Owned by".

---

Here, the model has successfully completed the fields for this tab in figure 51. Unable to find the next field because it is in the 'Financial' tab, the model fills the 'Serial Number' field with the 'Cost center' information. After this mistake, the model then finds how to change tabs, and does it thinking this tab is filled correctly:

---

**LLM Output**

```
<think>
I have successfully filled in the "Serial number" field with the value
"SN-566be2cf-3f63-4e26-8aa5-54ad26c07875".  The next field to fill
is the "Cost center" with the value "IT".  The "Cost center" field is
not visible in the current observation, so I need to navigate to the
appropriate tab where this field is located.  Based on the AXTree, the
"Financial" tab likely contains the "Cost center" field.  I will click
on the "Financial" tab to proceed.
</think>
<action>
click('a250')
</action>
```

---

The agent asserts that it has filled the 'Serial Number' field, but it does not take into account the fact that it overwrote it right after.

**Action Consequences Assessment - Repeated Actions**

Agents can sometimes fall into a loop, where they will retry the same action over and over. Typically, if a piece of information is missing on the page, the agents can sometimes start scrolling aimlessly over and over, trying find the missing data. This process can go on for the rest of a run, or randomly stop if the agent manages to get out of this mode. From the state in Fig. 53, the agent must go into the 'Hardware' section and buy a 'Loaner Laptop'.

At some point, it falls into a loop and only outputs the following action:

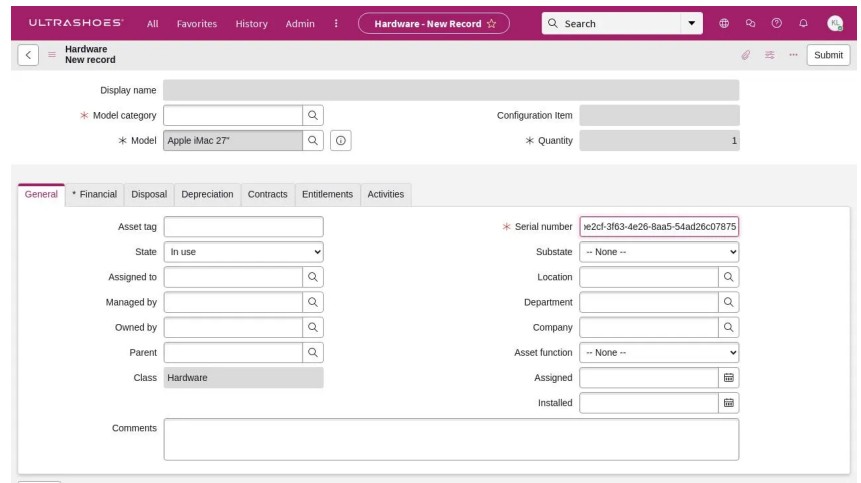

**Figure 51:** The agent successfully completed the 'Serial Number' field.

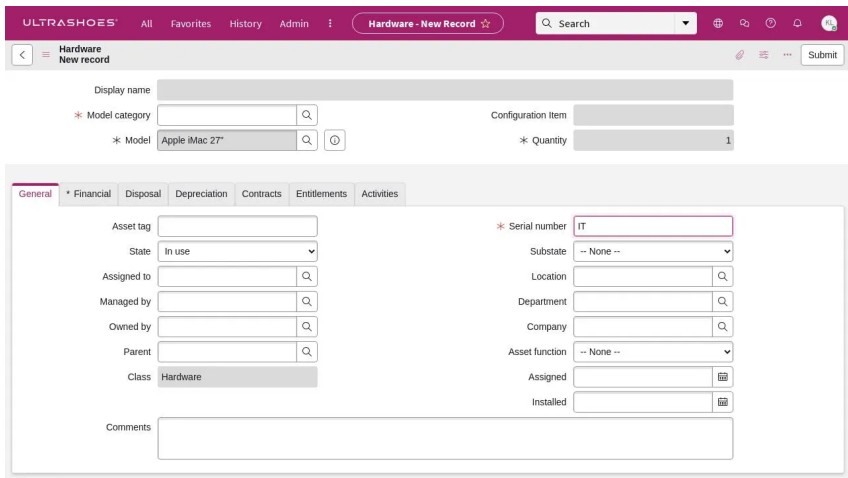

**Figure 52:** The agent overwrites the 'Serial Number' field with another field.

---

**LLM Output**

```
<think>
The previous attempts to click on the "Hardware" link and use the search
functionality have failed.  I have tried scrolling the page down to
make the "Hardware" link visible, but it seems that the element is still
not interactable.
I noticed that there is a "Search" combobox with bid '113' that is
visible and clickable.  I will attempt to click on this combobox to
focus on it and then type "Hardware" to search for the relevant section.
</think>
<action>
click('113')
</action>
```

In this example, bid 113 corresponds to the search bar. As clicking on it does not change the page or cause any error, the model keeps on repeating this action until the end of the episode.

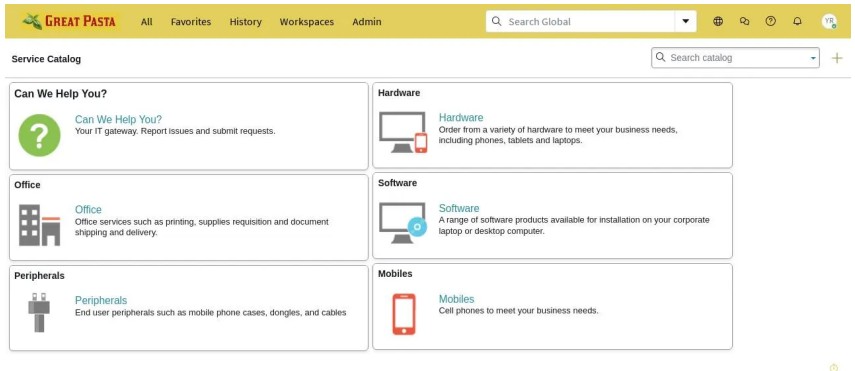

**Figure 53:** Service Catalog causing the model to loop over the same useless action

# G    Additional Discussion

## G.1    Limitation: Restricted to ServiceNow

The tasks in WorkArena++ are confined to ServiceNow and do not involve interacting with other software. It is thus reasonable to wonder: **(Q1)** whether insights from WorkArena++ are expected to generalize to other enterprise software environments and **(Q2)** if including more software environments would result in a benchmark that is more representative of general enterprise workflows. We hereby address those questions.

**Q1:** We believe that the answer is yes for three main reasons. First, our tasks are built around basic UI components (e.g., lists, forms, dashboards) that are ubiquitous across various enterprise software platforms (e.g., SalesForce, SAP). Second, as highlighted in our error analysis (§ 4.5), most failures are not tied to ServiceNow-specific issues. Instead, they stem from challenges like understanding the task goal, hallucinating actions, etc. Third, and most importantly, the tasks primarily evaluate complex reasoning and planning skills in agents, which are largely independent of the specific user interface.

**Q2:** Yes, definitely. However, we chose to limit the scope of the present benchmark to ServiceNow for the following reasons:

- WorkArena++ integrates into the broader BrowserGym [Drouin et al., 2024] ecosystem, which already contains benchmarks that evaluate cross-application performance (e.g., WebArena; Zhou et al. [2023]).
- ServiceNow Personal Developer Instances allow for a simple user experience, where the user does not have to launch a web server to run evaluations on the benchmark (as is common in related work).
- Tasks in WorkArena++ are compositions of atomic subtasks that involve interacting with common UI elements. Hence, the benchmark can easily be extended with tasks inspired by workflows that are relevant in other software environments.

That said, we acknowledge that creating multi-environment tasks is an exciting and promising direction for future work aimed at expanding this benchmark (e.g., to make an L4 set of tasks). For example, one could replace the built-in ServiceNow knowledge base in L3 tasks with an external one (e.g., Wikipedia). Realistically, all environments used in related work (e.g., WebArena) are within reach.

## G.2    Expanding the Benchmark – Ensuring Long-Term Relevance

How can WorkArena++ be updated or expanded to remain challenging and relevant? The modular design of WorkArena++ makes it easy to add new tasks. One can create complex workflows by composing "atomic tasks". Creating new "atomic tasks" simply involves writing setup, teardown,

oracle, and validation functions (as shown in Fig. 3). For an example of compositional task creation, please refer to the `GetWarrantyExpirationDateTask` task in the code base.

Moreover, there are many potential directions in which the benchmark could be expanded as the performance of agents increases. Examples include:

- **Longer and more complex workflows:** Inspiration could be drawn from the O*Net database [National Center for O*NET Development, 2024], which catalogs key tasks performed by job occupation (https://www.onetonline.org/).

- **Workflows that require collaboration:** This could include tasks where agents must delegate subtasks to colleagues (either agentic or human) or collaborate toward completion. One interesting observation is that the design of WorkArena++ makes it fairly easy to implement tasks where an agent must delegate work. One simply needs to run the "oracle function" for the atomic task (or trajectory) that was delegated by the agent. The agent could be penalized for too many delegations. The ServiceNow platform has collaboration features that would make the elaboration of such tasks feasible (e.g., discuss via chat, leave comments for one another on a ticket, etc.).

- **Workflows that involve interacting with multiple external software:** The number of such potential tasks is enormous. The main complexity is in interfacing with such software to implement setup, teardown, oracle, and validation functions (see Fig. 3).

We keep this for future work but welcome contributions from the open-source community.

### G.3 Context-size sensitivity

We ran an experiment to compare the impact of context-length on overall agent performance for WorkArena-L1. For this, we compared the performance of a Llama3-based agent using the full context (130k), 50% context (64k), 25% context (32k), and 10% context (13k). This experiment resulted in very similar performance across all different allowed context lengths. Even though this result is interesting, we would like to caution that similar performances across different context lengths should not be expected on benchmarks with more complex tasks, as these will likely require greater memory and detail in the prompts.

**Table 3:** Ablation study for **Llama3** on WorkArena-L1. Success rate$_{\pm \text{Standard error}}$ (SR $_{\pm \text{SE}}$).

| Context size | 130k | 64k | 32k | 13k |
|---|---|---|---|---|
| SR % $_{\pm \text{SE}}$ | 17.9 $_{\pm 2.1}$ | 14.8 $_{\pm 2.0}$ | 17.6 $_{\pm 2.1}$ | 18.2 $_{\pm 2.1}$ |

