# OpenReview forum: "WorkArena++: Towards Compositional Planning and Reasoning-based Common Knowledge Work Tasks"
_NeurIPS.cc/2024/Datasets_and_Benchmarks_Track — NeurIPS 2024 Track Datasets and Benchmarks Poster_

### Official Review · Reviewer_8Prt · 2024-07-17
**Excellent benchmark with thoughtful evaluation and analysis**

**Rating:** 8
**Confidence:** 4
**Correctness:** The design of the benchmark and evalu…
**Clarity:** The paper is exceptionally clear and …

**Review:**

######## Strengths ########

1. The benchmark is well-designed and carefully described in an appropriate level of detail
2. The empirical evaluation is excellent: it studies multiple LLM/VLM-based agents and humans, includes results on comparable benchmarks for comparison, and analyzes in detail the failure modes of the AI agents

######## Opportunities ########

My main concern with this work is that by introducing an open-source benchmark, easy to use for fine-tuning (by design) will quickly render the benchmark useless for benchmarking and turn it into a source of data for future LLMs.

######## Recommendation ########

I recommend that this paper is accepted for publication. The benchmark is well thought-out, carefully explained in a way that transmits the rationale behind each design choice, and the evaluations on the benchmark are well carried out. The experiments in Sec 4 enable extracting multiple insights, which are valuable toward understanding the current state of LLM-agents.

######## Arguments ########

The benchmark itself is carefully designed. It contains two levels of difficulty for every task, which enable highlighting the differences between failing to "plan" and failing to "execute". A method that performs well on L2 (where step-by-step instructions are provided) may be poor at constructing a sequence of steps, but is capable of executing all the individual steps in sequence; a method that performs well on L3 additionally needs to construct the sequence of steps necessary to attain a goal.

The benchmark further contains 5 types of skills to evaluate: planning, information retrieval, data-driven decision-making, memorization, and "infeasibility". It is clear from reading through Sec 3.2 that a lot of care went into designing this benchmark.

Overall, the benchmark is described in a great level of detail and clarity, including rationale for each relevant design choice. Some questions that I had while reading through Sec 3:
- "In addition to following these steps, succeeding at L2 tasks requires thinking, reasoning, and contextual understanding" --> What does this statement mean precisely? Maybe an example distinguishing the type of reasoning/thinking/context needed to solve L2 vs L3 tasks might be illustrative.
- Upon reading the example tasks from the planning skill, I wondered how L2 instructions would differ from L3 for these cases. For "redistributing work among emplyees based on occupancy", would L2 instructions specify which employee should be assigned to which task? Additional examples of the L2/L3 distinction across the various skills (potentially in an appendix, referenced in Sec 3.1) would be helpful. I believe part of this is included in Appendix C, but connecting that to the paper and using it to explain the L2/L3 distinction would be useful.

I found the experiments to be quite comprehensive and insightful. I especially appreciated the discussion in Sec 4.3 about the failure modes of the web-agents, which boils down to:
- Information retrieval errors: fail to retrieve relevant info or identify relevant elements of the page and act on wrong ones
- Exploration: struggle to open new tabs or expanding foldable elements
- Hallucination: sometimes hallucinate actions (eg imaginary buttons, ask for help)
- Goal understanding: thought process is correct but action is incorrect, or fail to comprehend subtasks (eg modify locked ticket values)
- Action consequences assessment: believe progress was made when not, or retry same action when no progress

The one relatively important concern that I have with this submission is that it is unclear how the benchmark will remain relevant over time as an evaluation tool.
- Sec 3.3: If benchmark is online, freely available, and even provides finetuning data, what's its use going to be once LLMs incorporate it in their training? Presumably they'll be able to solve these tasks (by potentially overfitting to them) but we won't know how well they work on tasks outside this benchmark.
- Sec 4: "valuable tool for evaluating and improving web agents" -- the challenge I see is that if web agents are improved by using the benchmark, then the benchmark would become useless for evaluating those agents.
- Sec 4.1: While it's good to see that the authors reserve some seeds for evaluation (though the wording "we encourage" suggests that this can't be enforced), this seems insufficient: who says that generalizing to unseen seeds implies generalizing to similar tasks outside the benchmark?

I encourage the authors to discuss this point in their manuscript.

**Strengths:**

1. The benchmark is well-designed and carefully described in an appropriate level of detail
2. The empirical evaluation is excellent: it studies multiple LLM/VLM-based agents and humans, includes results on comparable benchmarks for comparison, and analyzes in detail the failure modes of the AI agents

**Additional Feedback:**

The following points are provided as feedback to hopefully help better shape the submitted manuscript, but did not impact my recommendation in a major way.

Abstract
- Quite descriptive and sounds exciting

Sec 2
- So far everything is clear and well-motivated
- The background section is short but appears to give the necessary info to understand how the 2 main building blocks are used

Sec 4.2
- Observation space: the authors provide the history of actions and thoughts since the start of the episode. Does this mean that each action is treated as a standalone interaction with the environment, instead of one interaction within a sequence or "session"?

Sec 4.3
- Presumably the success rate is in percentage? This should be stated

Sec 4.4
- Are humans also given a max number of actions?

Typos/style/grammar
- There's a double period in line 217 (Sec 4.2)

**Documentation:**

The benchmark is well documented.

**Ethics:**

No.

**Limitations:**

Limitations are well-addressed in a standalone Section 7.

**Opportunities For Improvement:**

My main concern with this work is that by introducing an open-source benchmark, easy to use for fine-tuning (by design) will quickly render the benchmark useless for benchmarking and turn it into a source of data for future LLMs.

**Relation To Prior Work:**

Beside the actual benchmark, one of the useful insights of this work is that LLM agents fail to solve long-horizon problems autonomously. It might be worth highlighting prior work from the planning literature that has observed this phenomenon (e.g., see [1] and the references in that work).

[1] Kambhampati et al. "Position: LLMs Can’t Plan, But Can Help Planning in LLM-Modulo Frameworks." ICML 2024.

**Summary And Contributions:**

The submission introduces a benchmark for evaluating the ability of web agents (LLMs or VLMs) to execute typical enterprise workflow tasks. Each task requires executing a sequence of steps (each of which can be thought of as an atomic subtask) to achieve a goal. The goals can be specified as a set of sep-by-step instructions (in the easier L2 level) or as a higher-level overall goal (in the harder L3 level). The evaluation shows that web agents fail catastrophically in the benchmark, while humans excel at it. The authors then discuss the failure modes observed in the web-agent evaluations.

---

> ### Author Rebuttal · Authors · 2024-08-16
>
> Thank you for your positive assessment of our work. Please find a response to your comments and questions below.
>
> &nbsp;
>
> > My main concern with this work is that by introducing an open-source benchmark, easy to use for fine-tuning (by design) will quickly render the benchmark useless for benchmarking and turn it into a source of data for future LLMs.
>
> > Sec 3.3: If benchmark is online, freely available, and even provides finetuning data, what's its use going to be once LLMs incorporate it in their training? Presumably they'll be able to solve these tasks (by potentially overfitting to them) but we won't know how well they work on tasks outside this benchmark.
>
> > Sec 4: "valuable tool for evaluating and improving web agents" -- the challenge I see is that if web agents are improved by using the benchmark, then the benchmark would become useless for evaluating those agents.
>
> > Sec 4.1: While it's good to see that the authors reserve some seeds for evaluation (though the wording "we encourage" suggests that this can't be enforced), this seems insufficient: who says that generalizing to unseen seeds implies generalizing to similar tasks outside the benchmark?
>
> **Response**: Thank you for pointing these out. We agree and believe that they all encompass a common, and in-fact, a prevalent issue with open-source benchmarks. It is often difficult to control what people might do with our code and benchmark, and while we could have undisclosed task sets for evaluation, we believe that this might lead to added inertia for wider evaluations (e.g., significant resources required to evaluate contributed agents). Nevertheless, we strongly believe in the value of open-source benchmarks, many of which have historically been instrumental in advancing machine learning (e.g., ImageNet).
>
> Here are some measures we are taking to mitigate these issues. Please let us know if you have more ideas in this regard.
>
> - **Leaderboard**: We will host an open leaderboard and only include methods that stem from reproducible research, which we can inspect to ensure they do not make unintended use of the benchmark. One caveat here is that such inspection will not be possible for closed-source agentic systems.
> - **Non-hosted traces**: We will not host fine-tuning traces online to minimize the chances of inclusion into training sets acquired via scraping. Rather, traces must be generated by running a script in our code-base.
> - **Guidelines**: WorkArena++ is designed as an evaluation benchmark, primarily. We will ensure that the benchmark is accompanied by clear fine-tuning and evaluation guidelines to encourage proper adoption by the community.
> - **Expansion**: The progressive inclusion of new tasks over time could mitigate issues due to unintended fine-tuning. Performance might be good on earlier tasks, but significantly worse on new ones. This is an easy-to-implement mitigation.
>
> &nbsp;
>
> > "In addition to following these steps, succeeding at L2 tasks requires thinking, reasoning, and contextual understanding" --> What does this statement mean precisely? Maybe an example distinguishing the type of reasoning/thinking/context needed to solve L2 vs L3 tasks might be illustrative
>
> **Response**: Thank you for the suggestion. L2 and L3, both levels of tasks do require thinking, reasoning, planning, and contextual understanding abilities. This contrasts with the WorkArena-L1 tasks that are significantly low-level and whose goals are closely linked to the actions needed to complete them (filling a form, sorting a list’s columns, etc.). Even for solving an L2 task, the agent would simply be given high level steps, such as navigate to a page and perform a task, but would still need to figure out the exact steps to accomplish it, which does require all the above abilities listed. For example, the agent could be instructed to navigate to a dashboard, look at the inventory of products, and find all products whose stock levels are below the mean. In such a case, the agent wouldn’t be told that they need to hover over the charts to see the numerical values, nor how they should proceed to calculate a mean. They would need to figure out how to extract the information that is asked, use it to calculate mean, and then figure out products that have stocks below this value.
>
> To concretely distinguish between an L2 and L3 task, we consider the easy expense management task for example. The goal in L2 is displayed below. While the goal highlights all the necessary steps to complete the task, it does not describe how to accomplish them (e.g. how to use the menu to navigate)
>
> **L2 Goal**:
> ```
> Managing Your Existing Expenses
> Concretely, you need to complete the following steps:
> 1. Navigate to the "Expense Lines" module of the "Cost" application.
> 2. Create a filter for the list to extract all entries where:
>     - "Short description" contains "#SERIES-0cbf9a92-4"
> 3. Delete expense lines with duplicated short descriptions, keeping only one.
> ```
>
> For the same task, the L3 goal provided to the agent is simply:
> ```
> Please complete the following task.
> ```
> , accompanied by a ticket that says to refer to a KB article containing the rules for expense management. Therefore L3 requires the agent to understand it needs to solve the task from the starting web page, navigate to the KB article, remember the expense management rules and apply them, while in L2 the agent directly has the rule to apply.
>
> We would add more clearer distinctions and details with illustrative examples between the L2 and L3 tasks in section 3.1 for clarity.

---

> > ### Author Rebuttal · Authors · 2024-08-16
> >
> > ### Continuing from above...
> >
> > &nbsp;
> >
> > > Upon reading the example tasks from the planning skill, I wondered how L2 instructions would differ from L3 for these cases. For "redistributing work among employees based on occupancy", would L2 instructions specify which employee should be assigned to which task? Additional examples of the L2/L3 distinction across the various skills (potentially in an appendix, referenced in Sec 3.1) would be helpful. I believe part of this is included in Appendix C, but connecting that to the paper and using it to explain the L2/L3 distinction would be useful.
> >
> > **Response**: That is a good point and we will add a clearer connection to the appendix for more detail. The workload balancing task involves re-assigning a work item from the busiest user to the least busy user based on a dashboard. The L2 goal contains clear  information about where to navigate and how to filter the dashboard list to find the appropriate one, while the L3 task description simply tells the agent to refer to the work assignment protocol. It adds the additional complexity of having to retrieve the information on how to perform the work assignment instead of directly giving the rule -assigning from the busiest user to the least busy.
> >
> > &nbsp;
> >
> > > Observation space: the authors provide the history of actions and thoughts since the start of the episode. Does this mean that each action is treated as a standalone interaction with the environment, instead of one interaction within a sequence or "session"?
> >
> > **Response**: Yes, each action is a standalone interaction with the environment. We considered alternatives, like batches of actions, but the agents sometimes failed to consider the effect of their actions on the state of the page, resulting in poorer results on average.
> >
> > &nbsp;
> >
> > > Presumably the success rate is in percentage? This should be stated
> >
> > **Response**: Thanks for pointing this out. This is correct, and we will add this clearly in the text.
> >
> > &nbsp;
> >
> > > Are humans also given a max number of actions?
> >
> > **Response**:  No, we did not explicitly limit the number of actions that we allowed our human evaluators on our evaluation platform, mainly because of simplicity. In the case of AI agents, we do limit it not so as to impose a constraint on them, but due to resource availability (such as credits and rate limits). The number of steps we allow an AI agent to act for are sufficient to solve the tasks, and hence, even removing this constraint and allowing infinite steps would not lead to change in their performance. We will add this detail in the paper to prevent confusions, thank you.
> >
> > &nbsp;
> >
> > > There's a double period in line 217 (Sec 4.2)
> >
> > **Response**: Thank you!
> >
> > &nbsp;
> >
> > > Beside the actual benchmark, one of the useful insights of this work is that LLM agents fail to solve long-horizon problems autonomously. It might be worth highlighting prior work from the planning literature that has observed this phenomenon (e.g., see [1] and the references in that work).
> >
> > **Response**: Thank you for the pointer, we will add this in our related work and point it out in our analyses!

---

> > > ### Comment · Reviewer_8Prt · 2024-08-21
> > >
> > > Thank you for responding to each of my questions and concerns. Please be sure to include these in the final revision of your manuscript.

---

> > > > ### Author Response · Authors · 2024-08-21
> > > >
> > > > Thank you for your quick response and acknowledgement of our rebuttal. We will definitely include the changes in the revised version of the paper.

---

### Official Review · Reviewer_VC5R · 2024-07-24
**New Agent Benchmark for Enterprise Workflows**

**Rating:** 5
**Confidence:** 3
**Clarity:** The paper is clear, well written and …

**Review:**

Overall, this paper is clear, well-motivated, and provides a new benchmark for evaluating agent performance on realistic enterprise workflows using enterprise software. Such benchmarks for enterprise workflows are critical for the development of AI agents in enterprise settings, and potentially enable large efficiency gains from AI agents.

However, one major weakness of the paper is that despite the efforts increasing visual diversity of the environment and tasks, all enterprise workflow tasks are done exclusively on the ServiceNow platform and might not be representative of general tasks in enterprise workflows. Even though ServiceNow platform contains all the basic and rather diverse UI elements in typical enterprise web software, this paper could benefit from tasks in other enterprise software platforms (e.g. GitLab in WebArena) without adding more complex scenarios such as OS level / multi-tool use cases.

In addition, the major difference between L2 and L3 tasks is that L3 requires interpretation of the ServiceNow ticket description, and the operation to open / close tickets when done. It might be clearer to isolate the task of interpreting general, vague description into concrete smaller steps as its own benchmark, even though completing the L3 tasks are the ideal final states for enterprise agents.

**Strengths:**

- The paper is well-motivated and clearly written. With the race to capture value in enterprise AI agents, such benchmark can provide valuable contribution to both academic community and industry.
- The authors create complex multi-step tasks that mimic real-world enterprise scenarios, including the ideal end-to-end state of agents resolving service tickets.
- The authors have provided clear documentation on tasks, experiment setup and results, emphasized uniform setup, and offered robust evaluation guidelines.

**Additional Feedback:**

Currently I do not have additional feedback that are not covered in the sections above.

**Correctness:**

To the best of my knowledge, there are no correctness issue regarding benchmark evaluation methods and experiment design. I have also voiced my other concerns in the sections above.

**Documentation:**

The authors provided sufficient details, including code and GitHub repository to support reproducibility.

**Ethics:**

There are no further ethical concerns regarding this paper to the best of my knowledge.

**Limitations:**

In the main paper, the authors discussed the potential limitations in the task comprehensiveness, single platform, lack of security guardrails, models tested and the human learning effect from the human evaluation process. The authors have reasonably addressed most of the limitations in the paper.

**Opportunities For Improvement:**

The paper could benefit from discussions with regards to the following questions:

1. As mentioned before, all enterprise workflow tasks are done exclusively on the ServiceNow platform and might not be representative of general tasks in enterprise workflows. It might be beneficial to contain more diverse tasks in other enterprise workflow tools.
2. When looking at the demographics of human evaluators, around 3/4 have worked at ServiceNow, all of them with undergrad degrees, and half with advanced degrees. Is that representative of the general user demographics of ServiceNow? The high performance of human evaluators might not reveal the general cognitive complexity of the tasks for general first-time users (if the models are not fine-tuned and looking at zero-shot results only).
3. As mentioned above about the difference between L2 and L3 tasks - would it be clearer to isolate the task of interpreting general, vague description into concrete smaller steps as its own benchmark?
4. The authors have set up custom icons and different color schemes for the platform, but the general layout and access tree structure stays the same. Is there a way to increase visual diversity in terms of web layout?
5. Are there more details regarding the standard error for both agent and human evaluations - how many trials are done for each task? Are there multiple trials for each task, even if the task is performed successfully? There is only the mention of maximum 4 re-trys if the model is stuck.
6. Are there any attempts on prompt engineering, or the current results that existing models can hardly complete any L2 or L3 purely due to model capability limitation?
7. Given that human evaluators were given 15 minutes of exploration and 15 minutes of tutorial, how would that make a difference in performance with model finetuning or instruction finetuning?
8. The paper mentioned lack of exploration as a failure mode - can it be improved via prompting?
9. The paper mentioned prompt truncation - is context window a limiting factor in model performance? Models tested all have various context windows.
10. The paper mentioned the learning effect on human evaluators as they became more familiar and efficient with the software environment - which proves the point of adding more enterprise platforms into the benchmark. Are there other ways to isolate the effect through experiment setup?
11. The paper acknowledged the limitation that it does not address guardrails, consistency or robustness. Would it be possible to add task that tests agent performance consistency?

**Relation To Prior Work:**

To the best of my knowledge, this work discusses how it differs from previous contributions.

**Summary And Contributions:**

This paper introduces a new agent benchmark, WorkArena++, to evaluate web agents by mimicking workflows performed routinely by knowledge workers using enterprise software such as ServiceNow. WorkArena++ tests various complex skills of LLM and VLM-based agents including planning, decision-making, retrieval, mathematical reasoning, and the ability to identify infeasible tasks. The results reveal that state-of-the-art LLMs and VLMs fail in almost all tasks in the benchmark, while humans achieve very high performance.

---

> ### Author Rebuttal · Authors · 2024-08-16
>
> Thank you for your feedback. We are pleased that you acknowledge the relevance of the benchmark. Below is our point-by-point answer to your comments, which we hope will answer your concerns and questions.
>
> &nbsp;
>
> > As mentioned before, all enterprise workflow tasks are done exclusively on the ServiceNow platform and might not be representative of general tasks in enterprise workflows. It might be beneficial to contain more diverse tasks in other enterprise workflow tools.
>
> **Response**: These are valid concerns and this limitation is disclosed in our submission. Nevertheless, we believe that, despite this limitation, the present benchmark provides valuable and generalizable insights beyond ServiceNow. Please see our response in the “General Response” and do let us know if you would like to discuss this further.
>
> &nbsp;
>
> > When looking at the demographics of human evaluators, around 3/4 have worked at ServiceNow, all of them with undergrad degrees, and half with advanced degrees. Is that representative of the general user demographics of ServiceNow? The high performance of human evaluators might not reveal the general cognitive complexity of the tasks for general first-time users (if the models are not fine-tuned and looking at zero-shot results only).
>
> **Response**: Thank you, these are important questions and we will make sure to include the following comments in the revision of Appendix A.
>
> - **ServiceNow Employees**: Out of our evaluators, 11 out of 15 have been employed by ServiceNow. However, this number must not be interpreted as “people who have interacted with the product before”. In fact, many of these people have roles that do not involve interaction with the product. For example,  4 of these 11 evaluators were student researchers who, by the nature of their work, are not exposed to the product. Moreover, we emphasize that 46.7% of our evaluators declared being first-time users of the product, and 86.7% claim to use the product at most a few times per month. Finally, we stress that the tasks in WorkArena++ mostly evaluate planning and reasoning skills, beyond basic interaction with the product that some evaluators could be familiar with.
>
> - **Study degrees**: You rightly note that this bias is present in our pool of evaluators. Nevertheless, we believe it does not invalidate our conclusions, since none of our tasks involve skills that one would acquire through an advanced degree. Among the “general user demographics” of ServiceNow, all users must be able to manipulate basic UI components, such as forms, lists, etc. Beyond that, succeeding at our benchmark involves following a clear set of instructions and basic reasoning that should be well within reach for anyone with the ability to interact with the software. Hence, while this bias is present, we argue that it is not a significant driver of success on the benchmark.
>
> - **Zero-shot models**: Agents based on LLMs cannot be characterized as “first time users”. Our exploration revealed that these models have extensive knowledge of the ServiceNow platform, such as very detailed knowledge of available APIs, detailed information about the underlying data structure, and knowledge of how to solve certain tasks in the product. This likely comes from training on publicly available documentation and discussions in support forums. Hence, we believe that the relative lack of exposure to the platform of most human evaluators and the knowledge held by LLMs/VLMs helps in normalizing possible discrepancies in our human evaluation.
>
> &nbsp;
>
> > As mentioned above about the difference between L2 and L3 tasks - would it be clearer to isolate the task of interpreting general, vague description into concrete smaller steps as its own benchmark?
>
> **Response**: We would like to clarify that the major difference between L2 and L3 tasks is not that L3 requires an interpretation of vague descriptions. Rather, in L3, the agent receives its work through a ticketing system and must retrieve instructions from a knowledge base to understand how to proceed. However, all instructions are clearly stated and there is no missing information. This was meant as an increment over L2 tasks that is closer to how agents solve tasks in the real world. Nevertheless, the idea of designing tasks that are vague and require interpretation is an interesting one and would be a great addition to a future L4 set of tasks.
>
> &nbsp;
>
> > The authors have set up custom icons and different color schemes for the platform, but the general layout and access tree structure stays the same. Is there a way to increase visual diversity in terms of web layout?
>
> **Response**: Yes, we could, for example, change the sorting of columns in lists or change the position of fields in forms. However, we did not do this since all of our tasks rely on basic built-in forms and tables for which we thought it would be more realistic to preserve the natural ordering. Given that the benchmark is extremely challenging, we keep this added diversity for future work, but will clarify this in the paper and keep this in mind for future evolutions of the benchmark.

---

> > ### Author Rebuttal · Authors · 2024-08-16
> >
> > ### Continuing from above...
> >
> > &nbsp;
> >
> > > Are there more details regarding the standard error for both agent and human evaluations - how many trials are done for each task? Are there multiple trials for each task, even if the task is performed successfully? There is only the mention of maximum 4 re-trys if the model is stuck.
> >
> > **Response**: In our experiments, we create curricula for both agents and humans that ensure uniform evaluation across the task skills (rows of Table 1 in the L2/L3 sections). Each curriculum is obtained by randomly sampling across skills and then sampling pairs of (task, seed) within tasks that evaluate that skill; hence yes, there can be multiple trials for tasks. The agent curriculum used to produce Table 1 contains between 32 and 56 pairs per skill, reported beside skill labels in the table. Agents are then evaluated on these pairs and the results are aggregated by skill. Standard errors are then calculated according to the standard definition, i.e., std_deviation/sqrt(n) where n is the number of results being aggregated for a given skill and std_deviation is their standard deviation.
> >
> > Separately, the 4 retries that you mention are a component of our experiments, not the evaluation. Their purpose is to add robustness to errors in LLM generation. At inference, if a model produces an output that we fail to parse, we retry up to 4 times because we consider it a failure.
> >
> > We hope this clarifies your questions and will revise the paper to improve clarity.
> >
> > &nbsp;
> >
> > > Are there any attempts on prompt engineering, or the current results that existing models can hardly complete any L2 or L3 purely due to model capability limitation?
> >
> > **Response**: We follow the prompt template and its variations from the WorkArena paper, that had achieved state-of-the-art results on both WebArena and WorkArena. The WorkArena paper performed ablations on miniwob and the WorkArena-L1 tasks to find relevant prompt templates and variations to optimize their prompting method. On top of this, we also performed certain ablations on the relatively simpler L2 tasks (such as Nav-Do in Appendix C.4.1) with different variations of the prompt but observed almost no impact on the performance. We believe that the current results are on account of not just the limitations in the capability of the model but also the simple agentic system, which lacks advanced components such as RAG and memory that would further help improve the performance. We emphasize these reasons as drivers of poor performance in our error analysis (section 4.5) but believe that evaluating complex agentic systems is outside the scope of this work that introduces our benchmark, and hence we leave that as future work. We will address this point in the limitations section as well as future work sections since they would provide interesting directions for researchers working on AI agents.
> >
> > &nbsp;
> >
> > > Given that human evaluators were given 15 minutes of exploration and 15 minutes of tutorial, how would that make a difference in performance with model finetuning or instruction fine-tuning?
> >
> > **Response**: The tutorial given to human evaluators contains very high-level information about which UI components are included in tasks and how to use our human evaluation platform (see Figure 6). This in itself does not reveal information that helps to solve tasks. In the 15 minutes of self-exploration, the humans learn how to manipulate the UI components, a skill that the agents might indeed acquire through fine-tuning. However, as emphasized by our error analysis (Section 4.5), most failures of agents are due to mistakes in planning and reasoning, not basic UI manipulation. Furthermore, as previously explained, LLMs/VLMs have access to a wealth of information on ServiceNow products, potentially giving agents an advantage well beyond knowledge of basic UI manipulation. We will revise Appendix A to emphasize these points of discussion. As for finetuning, we believe it remains an interesting future direction to improve agent reasoning/planning skills, which is why we enable the generation of action/observation traces from the benchmark.
> >
> > &nbsp;
> >
> > > The paper mentioned lack of exploration as a failure mode - can it be improved via prompting?
> >
> > **Response**: We agree that one could design a prompt to increase exploration in the agent. However, exploration requirements are different across the tasks. For example, figuring out how to manipulate a UI component (e.g., an autocompletion field) requires different exploration skills than finding what page can be used to create a new user in the system. As such, prompts would need to be tailored for groups of tasks, which we want to avoid since we are after generalist agents. In that sense, we believe that research directions like guiding exploration through the retrieval of relevant documentation seem more promising to explore in future work.
> >
> > &nbsp;
> >
> > > The paper mentioned prompt truncation - is context window a limiting factor in model performance? Models tested all have various context windows.
> >
> > **Response**: Certainly, we do believe that context length is also a limiting factor in the model performance, especially since some axtrees go up to 40k tokens - and thus cannot be fitted into the prompt. We did try extending the context length for open-source models such as llama3-70b up to 16k (ie twice the original) but observed a significant dip in performance owing to the model having low-quality outputs and a lot of parsing errors, and hence, did not mention the results in the paper. We would experiment with the newer series of llama-3.1 models which have a much larger context length and present the results for that as well. Furthermore, we would run an ablation study on the context length of specific models to further gain crucial insights into this and add it to the revised version. Thanks for the input.

---

> > > ### Author Rebuttal · Authors · 2024-08-16
> > >
> > > ### Continuing from above...
> > >
> > > &nbsp;
> > >
> > > > The paper mentioned the learning effect on human evaluators as they became more familiar and efficient with the software environment - which proves the point of adding more enterprise platforms into the benchmark. Are there other ways to isolate the effect through experiment setup?
> > >
> > > **Response**: The humans may have become familiar with the UI components during the experiment. However, it is unlikely that they learned to solve any given task since most evaluators were not asked to perform the same task twice. Among our 15 evaluators, only 3 had curricula that included a repeated task, and, importantly, these tasks were always presented under another level (L2/L3) and a different seed. The significant difference between the presentation of goals and instructions in L2/L3 task acts as further mitigation.
> > >
> > > For full transparency, here are the curricula assigned to each evaluator. The 3 cases mentioned above are:
> > > - Evaluator 0: OnBoardUser L2/L3
> > > - Evaluator 9: DashboardRetrieveIncidentAndMeanLesserFilterAssetListTask L2/L3
> > > - Evaluator 12 EasyExpenseManagementSmallTask L2/L3
> > >
> > > We have attached a pdf (please find it at the end of this response) that outlines all the tasks assigned to each human evaluator for your reference.
> > >
> > > &nbsp;
> > >
> > > > The paper acknowledged the limitation that it does not address guardrails, consistency or robustness. Would it be possible to add task that tests agent performance consistency?
> > >
> > > **Response**: It would definitely be possible. The ideal would be to do this at the BrowserGym level, which would directly apply to all benchmarks that it encompasses. One could think of conducting attacks on the agent as tasks are being solved and evaluating failure/success rates in such a setting. We also briefly touch upon this in the future work section discussing “tasks for evaluating safety and cybersecurity around agents” and is the topic of ongoing research.

---

> > > > ### Author Response · Authors · 2024-08-28
> > > >
> > > > We hope that we were able to respond to your points satisfactorily. Please let us know if you have any further concerns. Thank you!

---

> ### Comment · Reviewer_VC5R · 2024-09-01
>
> I would like to thank the authors for their efforts explaining the improvements and therefore increased the score to 6 to reflect the changes.

---

> > ### Author Response · Authors · 2024-09-01
> >
> > Thank you very much for acknowledging and responding to our rebuttal.

---

### Official Review · Reviewer_ysFs · 2024-07-29
**Benchmark for Web Agents in Enterprise Settings**

**Rating:** 7
**Confidence:** 3

**Review:**

WorkArena++ represents a novel contribution to the field of web agent benchmarking. Its focus on complex, multi-step enterprise tasks and the introduction of different difficulty levels (L2 and L3) set it apart from existing benchmarks.

This work addresses a crucial gap in evaluating web agents' capabilities in realistic enterprise settings. The stark performance difference between humans and current AI models highlights important areas for improvement in AI research.

Pros:

Introduces a large-scale, challenging benchmark for web agents in enterprise settings.

Includes detailed error analysis to guide future research.

Demonstrates a significant performance gap between humans and current AI models.

Offers increased visual diversity and improved task isolation compared to previous benchmarks.

Evaluates multiple state-of-the-art models.


Cons:

Limited coverage of tasks within the enterprise software domain.

Focuses solely on the ServiceNow platform, potentially limiting generalizability.

Human evaluation was conducted with a smaller subset of tasks due to resource constraints.

**Strengths:**

The paper's strengths lie in its comprehensive approach to creating a challenging, realistic benchmark that addresses a significant gap in current web agent evaluation.

The paper conducts comprehensive evaluation of multiple state-of-the-art models, and highlights important areas for improvement in AI research, particularly in planning, reasoning, and context understanding.

It provides valuable insights for both researchers and practitioners in the field of AI and automation, while also considering the broader societal implications of advancing web agent capabilities.

**Additional Feedback:**

Discuss strategies for ensuring the long-term relevance of the benchmark as AI technologies evolve. How can WorkArena++ be updated or expanded to remain challenging and relevant?

While focusing on ServiceNow is valuable, consider discussing how the insights from WorkArena++ might generalize to other enterprise software platforms.

Given the ability to generate observation-action traces, it would be interesting to see results from fine-tuning experiments. How much can model performance improve with task-specific fine-tuning?

Have you considered how WorkArena++ could be used to evaluate multi-agent systems or collaborative AI-human teams?

What do you see as the most critical next steps in bridging the performance gap between AI models and humans on these complex enterprise tasks?

**Clarity:**

The paper is well-written, presenting its ideas and results clearly and effectively. Key ideas are well-articulated and accessible to readers familiar with the field.

Some readers might benefit from more detailed explanations of certain technical aspects, particularly regarding the ServiceNow platform for those unfamiliar with it.

**Correctness:**

The benchmark is constructed in a sound way, and the evaluation methods and experiment design are appropriate and correctly performed. The claims made in the paper are generally supported by the presented evidence, with appropriate acknowledgment of limitations.

**Documentation:**

The authors have addressed several key aspects related to dataset/benchmark documentation and reproducibility, The code is included in the supplementary materials with documentation.

**Ethics:**

No concern.

**Limitations:**

The authors have made an effort to address limitations and potential negative societal impacts of their work. Discussion of potential mitigation strategies for negative impacts can be desirable, although it is not required.

**Opportunities For Improvement:**

Although comprehensive, the benchmark is limited to tasks within the ServiceNow platform, potentially limiting its generalizability to other enterprise software environments.

**Relation To Prior Work:**

The paper does discuss how WorkArena++ differs from previous contributions, particularly in comparison to the original WorkArena benchmark and other existing web agent benchmarks.

A table or figure directly comparing features of WorkArena++ with other benchmarks could provide a clearer visualization of its unique aspects.

**Summary And Contributions:**

This paper introduces WorkArena++, a new benchmark for evaluating web agents' capabilities in performing complex knowledge work tasks in enterprise environments. The key contributions include:

A set of 682 tasks that test advanced skills like planning, problem-solving, data-driven decision making, and information retrieval. These tasks are more complex and realistic compared to previous benchmarks.

Two levels of difficulty (L2 and L3) for each task, with L3 being more challenging as it requires agents to understand goals from ticket descriptions and retrieve information from knowledge bases.

An evaluation of state-of-the-art language and vision-language models on the benchmark, showing that current agents struggle to solve these tasks while humans perform well.

A detailed error analysis highlighting the current limitations of web agents in handling complex enterprise tasks.

The authors position WorkArena++ as a valuable tool for driving advancements in web agent capabilities, particularly for enterprise applications.

---

> ### Author Rebuttal · Authors · 2024-08-16
>
> Thank you for your positive assessment of our work. Please find a response to your comments and questions below.
>
> &nbsp;
>
> > Although comprehensive, the benchmark is limited to tasks within the ServiceNow platform, potentially limiting its generalizability to other enterprise software environments
> > Limited coverage of tasks within the enterprise software domain
> > Focuses solely on the ServiceNow platform, potentially limiting generalizability.
>
> **Response**:  Please see our response in the “General Response” and do let us know if you would like us to expand on some aspects. We will revise the paper accordingly.
>
> &nbsp;
>
> > Human evaluation was conducted with a smaller subset of tasks due to resource constraints.
>
> **Response**: This choice was made because the tasks in WorkArena++ are detailed and complex, requiring significant time and concentration to solve. However, we ensured that the “Human Curriculum” was representative of the “Agent Curriculum”, so performance on one would be informative of the other (see Appendix A for details). The code used to generate each of these curricula is available in our [codebase](https://github.com/ServiceNow/WorkArena/blob/6c13c7e35c818da3523e7ceeff989891f874111f/src/browsergym/workarena/__init__.py#L99).
>
> Additionally, to avoid any misinterpretation of the results, we separated the “Agent Curriculum” and “Human Curriculum” in Table 1. We hope this provides more context for our choices. Please let us know if you would like additional details included in the manuscript.
>
> &nbsp;
>
> > Some readers might benefit from more detailed explanations of certain technical aspects, particularly regarding the ServiceNow platform for those unfamiliar with it.
>
> **Response**: Thank you for the suggestion. We will provide more details about the ServiceNow platform by adding a subsection in the Background section before 2.1 and also refer the interested readers to additional documentation available online, particularly details relevant to WorkArena++.
>
> &nbsp;
>
> > A table or figure directly comparing features of WorkArena++ with other benchmarks could provide a clearer visualization of its unique aspects.
>
> **Response**: Thank you for the suggestion. We will produce such a table and include it in the revised manuscript. In the pdf attached (please see bottom of the message, after the response), we include a draft that we will expand on (e.g., adding other related benchmarks like visual-webarena). Please let us know if other relevant points of comparison come to mind.
>
> &nbsp;
>
> > Discuss strategies for ensuring the long-term relevance of the benchmark as AI technologies evolve. How can WorkArena++ be updated or expanded to remain challenging and relevant?
>
> **Response**: The modular design of WorkArena++ makes it easy to add new tasks. One can create complex workflows by composing “atomic tasks”. Creating new “atomic tasks” simply involves writing setup, teardown, oracle, and validation functions (as shown in Figure 3). As an example of compositional task creation, you can take a look at how the “GetWarrantyExpirationDateTask” is created in our [codebase](https://github.com/ServiceNow/WorkArena/blob/6c13c7e35c818da3523e7ceeff989891f874111f/src/browsergym/workarena/tasks/compositional/warranty_check.py#L86).
>
> Moreover, we have a number of ideas to expand the benchmark as performance increases:
> - Longer and more complex workflows. Inspiration could be drawn from the O*Net database, which catalogs the key tasks performed by job occupation ([https://www.onetonline.org/](https://www.onetonline.org/)).
> - Workflows that require communication (or delegation of work) between multiple agents or between agents and humans. The ServiceNow platform has collaboration features that would make the elaboration of such tasks feasible.
> - Workflows that involve interacting with multiple external software (as you suggest). The number of such tasks that could be created is enormous.
>
> We’d be glad to add a comment about such directions in the revised manuscript. Please let us know if other ideas come to mind.
>
> &nbsp;
>
> > Given the ability to generate observation-action traces, it would be interesting to see results from fine-tuning experiments. How much can model performance improve with task-specific fine-tuning?
>
> **Response**: Yes, we agree that this would be interesting to explore. Task-specific fine-tuning would likely improve performance on a given specific task, but would it enable good performance on other tasks? A number of questions come to mind. For example, would fine-tuning on atomic tasks (L1) lead to success in the longer compositional tasks (L2, L3)? Are there auxiliary tasks that could unlock success on the benchmark? Given the number of directions that could be explored, we believe this mandates careful exploration in a subsequent work. In the present work, we restrict the scope to designing the benchmark and consider the ability to generate such traces as an added bonus over the initial version of WorkArena. We will clarify this in our revision.

---

> > ### Author Rebuttal · Authors · 2024-08-16
> >
> > ### Continuing from above...
> >
> > &nbsp;
> >
> > > Have you considered how WorkArena++ could be used to evaluate multi-agent systems or collaborative AI-human teams?
> >
> > **Response**: Yes, this is a very good question. The ServiceNow platform includes a number of features that allow agents to collaboratively work on tasks (e.g., discuss via chat, leave comments for one another on a ticket, etc.). In future work, we intend to add tasks where agents must delegate subtasks to colleagues (either agentic or human) or collaborate toward completion. One interesting observation is that the design of WorkArena++ makes it fairly easy to implement tasks where an agent must delegate work. One simply needs to run the “oracle function” for the atomic task (or trajectory) that was delegated by the agent. The agent could be penalized for too many delegations. We are actively thinking about such extensions.
> >
> > &nbsp;
> >
> > > What do you see as the most critical next steps in bridging the performance gap between AI models and humans on these complex enterprise tasks?
> >
> > **Response**: Drawing insights from our error analysis in section 4.5, we believe the most important features that current agents lack are 1) a good memory management system to store and retrieve from and 2) a mechanism to bridge the gap between natural language actions and low level actions expressed through code. In many cases, we observed that agents design sound high-level plans but fail to translate them to the right actions with the right elements of the web pages. Moreover, the agents frequently remain stuck when attempting to interact with some elements they think are the right ones. One pattern that occurs frequently is the failure in navigation between sub-tasks. In this failure mode, the agent needs to use the menu to navigate to the page containing its next sub-task, but fails by cycling between a variety of actions that are not working.
> >
> > Other next steps include:
> > - Reducing the size of observations: The massive size of observations leads to costly and slow execution. Can it be reduced by relying on visual observations (like screenshots) or on subsets of the page DOM/accessibility trees obtained through goal-driven retrieval?
> > - Trace memories: Given a concise representation of observations, could an agent maintain a memory of traces, essentially learning to interact with complex UIs over time? This might alleviate issues where the agent get stuck in a loop of incorrect actions. It might even be possible to include a self-play aspect to this.
> > - Multi-agent systems: Can lower-level agents be specialized to interacting with some pages or UI components (e.g., a form-filling agent, a dashboard-reading agent), acting on behalf of a higher-level agent that produces a plan?
> > - Fine-tuning: This is likely to help, but careful exploration of which kind of fine-tuning traces are required remains to be done.

---

> > ### Comment · Reviewer_ysFs · 2024-08-24
> >
> > I acknowledge the rebuttal.

---

### Author Rebuttal · Authors · 2024-08-16

## General Response to the Reviewers

We sincerely thank the reviewers for their thoughtful feedback and careful evaluation of our work. We are encouraged by the positive tone of the reviews, which acknowledge that the benchmark is large-scale, challenging, and realistic, and can serve as a valuable tool for the development of enterprise AI agents. We are particularly pleased that the reviewers appreciated the clarity of the exposition, the soundness of our empirical evaluations, and the relevance of our benchmark to both the academic and industrial research communities.
Below, we address common comments in a general response and then provide detailed responses to each reviewer’s comments. We hope that our responses will address all concerns and look forward to further discussion.

&nbsp;

### Common questions/comments

> The benchmark is limited to the ServiceNow platform

**Response**: The reviewers rightly note that WorkArena++ is currently limited to tasks within the ServiceNow platform, a limitation that we acknowledged in the original submission. Their concerns center on two main questions, which we answer below.

1. Can we expect insights from WorkArena++ to generalize to other enterprise software environments?

$\quad$ Yes, we believe so for three main reasons. First, our tasks are built around basic UI components (e.g., lists, forms, dashboards) that are ubiquitous across various enterprise software platforms (e.g., SalesForce, SAP), as noted by VC5R. Second, as highlighted in our error analysis (Section 4.5), most failures are not tied to ServiceNow-specific issues. Instead, they stem from challenges like understanding the task goal, hallucinating actions, etc. Third, and most importantly, the tasks primarily evaluate complex reasoning and planning skills in agents, which are largely independent of the specific user interface.

2. Would including more software environments result in a benchmark that is more representative of general tasks in enterprise workflows?

$\quad$ Yes, definitely. However, we chose to limit the scope of the present benchmark to ServiceNow for the following reasons:

- WorkArena++ integrates into the broader BrowserGym ecosystem, which already contains benchmarks that evaluate cross-application performance (e.g., WebArena).
- ServiceNow Personal Developer Instances allow for a simple user experience, where the user does not have to launch a web server to run evaluations on the benchmark (as is common in related work).
- Tasks in WorkArena++ are compositions of atomic subtasks that involve interacting with common UI elements. Hence, the benchmark can easily be extended with tasks inspired by workflows that are relevant in other software environments.

$\quad$ That said, we acknowledge that creating multi-environment tasks is an exciting and promising direction for an L4 set. For example, one could replace the built-in ServiceNow knowledge base in L3 tasks with an external one (e.g., Wikipedia). Realistically, all environments used in related work (e.g., WebArena) are within reach.

$\quad$ If the reviewers agree, we will incorporate these reflections in our revised paper.


> Limited coverage of tasks within the enterprise software domain

**Response**: We agree that this is a limitation of our work, as acknowledged in the “Limitations” section of our submission. However, for the reasons outlined above, we believe that the current benchmark still provides valuable insights to the research community. Additionally, we would like to emphasize that WorkArena++ is open source and designed with compositionality in mind, allowing anyone to contribute new tasks to extend coverage in specific application areas. To encourage this, we will emphasize in the discussion that we welcome and encourage community contributions to further extend the scope of the benchmark.

---

### Decision · Program_Chairs · 2024-09-26

**Decision:**

Accept (Poster)

**Comment:**

This paper provides a large benchmark with realistic enterprise tasks that include a mix of planning, memorization, retrieval etc competences. While there was some concern that the workflows were all from ServiceNow platforms, the reviewers were generally positive about the contributions of the paper. There was also a suggestion from one of the reviewers that the paper make better connections to relevant work on planning capabilities of LLMs.

On the whole, the reviews were thoughtful and high quality and the authors provided comprehensive rebuttals that seemed to satisfy the reviewers (one of them made an official statement that they are increasing the score from 5 to 6 but seemed to have forgotten to do so; taking that as their  intent the paper has 8,7,6 as the scores).

I recommend acceptance.